# NAG-GS: Semi-Implicit, Accelerated and Robust Stochastic Optimizer

## Abstract

Classical machine learning models such as deep neural networks are usually trained by using Stochastic Gradient Descent-based (SGD) algorithms. The classical SGD can be interpreted as a discretization of the stochastic gradient flow. In this paper we propose a novel, robust and accelerated stochastic optimizer that relies on two key elements: (1) an accelerated Nesterov-like Stochastic Differential Equation (SDE) and (2) its semi-implicit Gauss-Seidel type discretization. The convergence and stability of the obtained method, referred to as NAG-GS, are first studied extensively in the case of the minimization of a quadratic function. This analysis allows us to come up with an optimal learning rate in terms of the convergence rate while ensuring the stability of NAG-GS. This is achieved by the careful analysis of the spectral radius of the iteration matrix and the covariance matrix at stationarity with respect to all hyperparameters of our method. Further, we show that NAG-GS is competitive with state-of-the-art methods such as momentum SGD with weight decay and AdamW for the training of machine learning models such as the logistic regression model, the residual networks models on standard computer vision datasets, Transformers in the frame of the GLUE benchmark and the recent Vision Transformers.

## 1 Introduction

Nowadays, machine learning, and more particularly deep learning, has achieved promising results on a wide spectrum of AI application domains. In order to process large amounts of data, most competitive approaches rely on the use of deep neural networks. Such models require to be trained and the process of training usually corresponds to solving a complex optimization problem. The development of fast methods is urgently needed to speed up the learning process and obtain efficiently trained models. In this paper, we introduce a new optimization framework for solving such problems.
**Main contributions of our paper:**

- We propose a new accelerated gradient method of Nesterov type for convex and non-convex stochastic optimization based on the Gauss-Seidel discretization;

- We analyze the properties of the proposed method both theoretically for the quadratic case and empirically on large variety of optimization problems;

- We show that our method is robust to the selection of learning rate values, memory-efficient compared with AdamW and competitive with baseline methods in various benchmarks.

**Organization of our paper:**

- Section 1.1 gives the theoretical background for our method.

- In Section 2, we propose an accelerated system of Stochastic Differential Equations (SDE) and a corresponding solver based on a specific discretization method. This method, called NAG-GS (Nesterov Accelerated Gradient with Gauss-Seidel Splitting), is initially discussed in terms of convergence for quadratic functions. Additionally, we apply NAG-GS to solve a 1-dimensional non-convex SDE and provide strong numerical evidence of its superior acceleration compared to classical SDE solvers in Section 2 of the supplementary materials.

- In Section 3, NAG-GS is tested to tackle stochastic optimization problems of increasing complexity and dimension, starting from the logistic regression model to the training of large machine learning models such as ResNet-20, VGG-11 and Transformers.

## 1.1 PRELIMINARIES

We start here with some general considerations in the deterministic setting for obtaining accelerated Ordinary Differential Equations (ODE) that will be extended in the stochastic setting in Section 2.1. We consider iterative methods for solving the unconstrained minimization problem:

$$\min_{x \in V} f(x), \tag{1}$$

where $V$ is a Hilbert space, and $f : V \to \mathbb{R} \cup \{+\infty\}$ is a properly closed convex extended real-valued function. In the following, for simplicity, we shall consider the particular case of $\mathbb{R}^n$ for $V$ and consider function $f$ smooth on the entire space. We also suppose $V$ is equipped with the canonical inner product $\langle x, y \rangle = \sum_{i=1}^n x_i y_i$ and the correspondingly induced norm $\|x\| = \sqrt{\langle x, x \rangle}$. Finally, we will consider in this section the class of functions $\mathcal{S}_{L,\mu}^{1,1}$ which stands for the set of strongly convex functions of parameter $\mu > 0$ with Lipschitz-continuous gradients of constant $L > 0$. For such class of functions, it is well-known that the global minimizer exists uniquely Nesterov (2018). One well-known approach to deriving the Gradient Descent (GD) method is discretizing the so-called gradient flow:

$$\dot{x}(t) = -\nabla f(x(t)), \quad t > 0. \tag{2}$$

The simplest forward (explicit) Euler method with step size $\alpha_k > 0$ leads to the GD method

$$x_{k+1} \leftarrow x_k - \alpha_k \nabla f(x_k).$$

In the field of numerical analysis, it is widely recognized that this method is conditionally $A$-stable. Moreover, when considering $f \in \mathcal{S}_{L,\mu}^{1,1}$ with $0 \le \mu \le L \le \infty$, the utilization of a step size $\alpha_k = 1/L$ leads to a linear convergence rate. It is important to highlight that the highest rate of convergence is attained when $\alpha_k = \frac{2}{\mu+L}$. In such a scenario, we have $\|x_k - x^\star\|^2 \le \left(\frac{Q_f-1}{Q_f+1}\right)^{2k} \|x_0 - x^\star\|^2$,

where $Q_f$ is defined as $Q_f = \frac{L}{\mu}$ and is commonly referred to as the condition number of function $f$ Nesterov (2018). Another approach that can be considered is the backward (implicit) Euler method, which is represented as:

$$x_{k+1} \leftarrow x_k - \alpha_k \nabla f(x_{k+1}), \tag{3}$$

This method is unconditionally $A$-stable. In a nutshell, A-stability in numerical ordinary differential equations characterizes a method's performance in the asymptotic regime, as time approaches infinity. An unconditionally A-stable method is one where the integration step can be arbitrarily large, yet the global error of the method converges to zero. We give more details about the notion in Appendix 1.3. Here-under, we summarize the methodology proposed by Luo & Chen (2021) to come up with a general family of accelerated gradient flows by focusing on the following simple problem:

$$\min_{x \in \mathbb{R}^n} f(x) = \frac{1}{2} x^T A x \tag{4}$$

for which the gradient flow in equation 2 reads simply as:

$$\dot{x}(t) = -A x(t), \quad t > 0, \tag{5}$$

where $A$ is a $n$-by-$n$ symmetric positive semi-definite matrix ensuring that $f \in \mathcal{S}_{L,\mu}^{1,1}$ where $\mu$ and $L$ respectively correspond to the minimum and maximum eigenvalues of matrix $A$, which are real and positive by hypothesis. Instead of directly resolving equation 5, authors of Luo & Chen (2021) opted to address a general linear ODE system as follows:

$$\dot{y}(t) = G y(t), \quad t > 0. \tag{6}$$

The main concept is to search for a system equation 6 with an asymmetric block matrix $G$ that transforms the spectrum of $A$ from the real line to the complex plane, reducing the condition number from $\kappa(A) = \frac{L}{\mu}$ to $\kappa(G) = O\left(\sqrt{\frac{L}{\mu}}\right)$. Subsequently, accelerated gradient methods can be

constructed from $A$-stable methods to solve equation 6 with a significantly larger step size, improving the contraction rate from $O\left(\left(\frac{Q_f-1}{Q_f+1}\right)^{2k}\right)$ to $O\left(\left(\frac{\sqrt{Q_f}-1}{\sqrt{Q_f}+1}\right)^{2k}\right)$. Moreover, to handle the convex case $\mu = 0$, the authors in Luo & Chen (2021) combine the transformation idea with a suitable time scaling technique. In this paper we consider one transformation that relies on the embedding of $A$ into some $2 \times 2$ block matrix $G$ with a rotation built-in Luo & Chen (2021):

$$G_{NAG} = \begin{bmatrix} -I & I \\ \mu/\gamma - A/\gamma & -\mu/\gamma I \end{bmatrix} \tag{7}$$

where $\gamma$ is a positive time scaling factor that satisfies

$$\dot{\gamma}(t) = \mu - \gamma(t), \quad \gamma(0) = \gamma_0 > 0. \tag{8}$$

Note that, given $A$ positive definite, we can easily show that for the considered transformation, we have that $\mathcal{R}(\lambda) < 0$, that is the real part of $\lambda$ is strictly negative, and this for all $\lambda \in \sigma(G)$ with $\sigma(G)$ denotes the spectrum of $G$, i.e. the set of all eigenvalues of $G$. Further, we will denote by $\rho(G) := \max_{\lambda \in \sigma(G)} |\lambda|$ the spectral radius of matrix $G$. Let us now consider the NAG block Matrix and let $y = (x, v)$, the dynamical system given in equation 6 with $y(0) = y_0 \in \mathbb{R}^{2n}$ reads:

$$\begin{aligned} \frac{dx}{dt} &= v - x, \\ \frac{dv}{dt} &= \frac{\mu}{\gamma}(x - v) - \frac{1}{\gamma}Ax \end{aligned} \tag{9}$$

with initial conditions $x(0) = x_0$ and $v(0) = v_0$. Before going further, let us remark that this linear ODE can be expressed as the following second-order ODE by eliminating $v$:

$$\gamma \ddot{x} + (\gamma + \mu)\dot{x} + Ax = 0, \tag{10}$$

where $Ax$ is therefore the gradient of $f$ w.r.t. $x$. Thus, one could generalize this approach for any function $f \in \mathcal{S}_{L,\mu}^{1,1}$ by replacing $Ax$ by $\nabla f(x)$, respectively, within equation 7, equation 9 and equation 10. Finally, some additional and useful insights are discussed in supplementary materials, Section 1.

## 2 MODEL AND THEORY

### 2.1 ACCELERATED STOCHASTIC GRADIENT FLOW

In the previous section, we presented a family of accelerated Gradient flows obtained by an appropriate spectral transformation $G$ of matrix $A$, see equation 9. One can observe the presence of a gradient term of the smooth function $f(x)$ at $x$ in the second differential equation equation 10. Let us recall that $Ax$ can be replaced by $\nabla f(x)$ for any function $f \in \mathcal{S}_{L,\mu}^{1,1}$. In the frame of this paper, function $f(x)$ may correspond to some loss function used to train neural networks. For such a setting, we assume that the gradient input $\nabla f(x)$ is contaminated by noise due to a finite-sample estimate of the gradient. The study of accelerated Gradient flows is now adapted to include and model the effect of the noise; to achieve this we consider the dynamics given in equation 6 perturbed by a general martingale process. This leads us to consider the following Accelerated Stochastic Gradient (ASG) flows:

$$\begin{aligned} \frac{dx}{dt} &= v - x, \\ \frac{dv}{dt} &= \frac{\mu}{\gamma}(x - v) - \frac{1}{\gamma}Ax + \frac{dZ}{dt}, \end{aligned} \tag{11}$$

which corresponds to an (Accelerated) system of SDE's, where $Z(t)$ is a continuous Ito martingale. We assume that $Z(t)$ has the simple expression $dZ = \sigma dW$, where $W = (W_1, ..., W_n)$ is a standard $n$-dimensional Brownian Motion. As a simple and first approach, we consider the volatility parameter $\sigma$ constant. In the next section, we present the discretizations considered for ASG flows given in equation 11.

## 2.2 Discretization: Gauss-Seidel Splitting and Semi-Implicitness

In this section, we present the main strategy to discretize the Accelerated SDE's system from equation 11. The main motivation behind the discretization method is to derive integration schemes that are, in the best case, unconditionally $A$-stable or conditionally $A$-stable with the highest possible integration step. In the classical terminology of (discrete) optimization methods, this value ensures convergence of the obtained methods with the largest possible step size and consequently improves the contraction rate (or the rate of convergence). In Section 1.1, we have briefly recalled that the most well-known unconditionally $A$-stable scheme was the backward Euler method (see equation 3), which is an implicit method and hence can achieve faster convergence rate. However, this requires to either solve a linear system or, in the case of a general convex function, to compute the root of a non-linear equation, both situations leading to a high computational cost. This is the main reason why few implicit schemes are used in practice for solving high-dimensional optimization problems. But still, it is expected that an explicit scheme closer to the implicit Euler method will have good stability with a larger step size than the one offered by a forward Euler method. Furthermore, assuming a Gaussian noise process, proposing a solver capable of handling a broad range of step size values is crucial. Specifically, allowing for a larger ratio $\alpha/b$ (with $b$ as the mini-batch size) increases the likelihood of converging to wider local minima, ultimately enhancing the generalization performance of the trained model, see Section 1 of supplementary materials for additional details on that matter. Motivated by the Gauss–Seidel (GS) method for solving linear systems, we consider the matrix splitting $G = M + N$ with $M$ being the lower triangular part of $G$ and $N = G - M$, we propose the following Gauss-Seidel splitting scheme for equation 6 perturbated with noise:

$$\frac{y_{k+1} - y_k}{\alpha_k} = M y_{k+1} + N y_k + \begin{bmatrix} 0 \\ \sigma \frac{W_{k+1} - W_k}{\alpha_k} \end{bmatrix} \tag{12}$$

which for $G = G_{NAG}$ (see (7)), gives the following semi-implicit scheme with step size $\alpha_k > 0$:

$$\begin{aligned} \frac{x_{k+1} - x_k}{\alpha_k} &= v_k - x_{k+1}, \\ \frac{v_{k+1} - v_k}{\alpha_k} &= \frac{\mu}{\gamma_k}(x_{k+1} - v_{k+1}) - \frac{1}{\gamma_k} A x_{k+1} + \sigma \frac{W_{k+1} - W_k}{\alpha_k}. \end{aligned} \tag{13}$$

Note that due to the properties of Brownian motion, we can simulate its values at the selected points by: $W_{k+1} = W_k + \Delta W_k$, where $\Delta W_k$ are independent random variables with distribution $\mathcal{N}(0, \alpha_k)$. Furthermore, ODE (8) corresponding to the parameter $\gamma$ is also discretized implicitly:

$$\frac{\gamma_{k+1} - \gamma_k}{\alpha_k} = \mu - \gamma_{k+1}, \quad \gamma_0 > 0. \tag{14}$$

As already mentioned earlier, heuristically, for general $f \in \mathcal{S}_{L,\mu}^{1,1}$ with $\mu \geq 0$, we just replace $Ax$ in equation 13 with $\nabla f(x)$ and obtain the following NAG-GS scheme:

$$\begin{aligned} \frac{x_{k+1} - x_k}{\alpha_k} &= v_k - x_{k+1}, \\ \frac{v_{k+1} - v_k}{\alpha_k} &= \frac{\mu}{\gamma_k}(x_{k+1} - v_{k+1}) - \frac{1}{\gamma_k} \nabla f(x_{k+1}) + \\ &\quad + \sigma \frac{W_{k+1} - W_k}{\alpha_k}. \end{aligned} \tag{15}$$

Finally, we introduce a method called the NAG-GS method (see Algorithm 1). In this method, we take into account the presence of unknown noise when computing the gradient $\nabla f(x_{k+1})$. We denote this noisy gradient as $\nabla \tilde{f}(x_{k+1})$ in Algorithm 1. Notably, in order to achieve strict equivalence with the scheme described in Equation (15), we have the relationship $\nabla \tilde{f}(x_{k+1}) = \nabla f(x_{k+1}) + \sigma \mu (1 - \frac{1}{b_k})(W_{k+1} - W_k)$, where $b_k$ is defined as $b_k := \alpha_k \mu (\alpha_k \mu + \gamma_{k+1})^{-1}$.

**Remark 1** (Complexity of NAG-GS algorithm compared to AdamW). *According to Algorithm 1, NAG-GS algorithm requires one auxiliary vector that matches the dimension of the trained parameters. In contrast, AdamW requires two auxiliary vectors of the same dimension. Hence, NAG-GS is expected to be more efficient than AdamW due to its lower computational complexity and memory requirements, enabling faster training and improving scalability for optimizing deep learning models with large datasets and resource-constrained environments.*

---

**Algorithm 1** Nesterov Accelerated Gradients with Gauss–Seidel splitting (NAG-GS).

---

**Input:** Choose point $x_0 \in \mathbb{R}^n$, some $\mu \geq 0, \gamma_0 > 0$.
   Set $v_0 := x_0$.
   **for** $k = 1, 2, \ldots$ **do**
      Choose step size $\alpha_k > 0$.
      ▷ Update parameters and state $x$:
      Set $a_k := \alpha_k(\alpha_k + 1)^{-1}$.
      Set $\gamma_{k+1} := (1 - a_k)\gamma_k + a_k\mu$.
      Set $x_{k+1} := (1 - a_k)x_k + a_k v_k$.
      ▷ Update state $v$:
      Set $b_k := \alpha_k\mu(\alpha_k\mu + \gamma_{k+1})^{-1}$.
      Set $v_{k+1} := (1 - b_k)v_k + b_k x_{k+1} - \mu^{-1}b_k \nabla \tilde{f}(x_{k+1})$.
   **end for**

---

Moreover, the step size update can be performed with different strategies, for instance, one may choose the method proposed by Nesterov (Nesterov, 2018, Method 2.2.7) which specifies to compute $\alpha_k \in (0, 1)$ such that $L\alpha_k^2 = (1 - \alpha_k)\gamma_k + \alpha_k\mu$. Note that for $\gamma_0 = \mu$, hence the sequences $\gamma_k = \mu$ and $\alpha_k = \sqrt{\frac{\mu}{L}}$ for all $k \geq 0$. In Section 2.3, we discuss how to compute the step size for Algorithm 1.

Let us mention that full-implicit discretizations have been considered and studied by the authors, these will be briefly discussed in supplementary materials, Section 1.2. However, their interests are, at the moment, limited for ML applications since the obtained implicit schemes use second-order information about $f$, such schemes are typically intractable for real-life ML models.

## 2.3 CONVERGENCE ANALYSIS OF QUADRATIC CASE

We propose to study how to select a maximum step size that ensures an optimal contraction rate while guaranteeing the convergence, or the stability of NAG-GS method once used to solve SDE's system 11. Ultimately, we show that the choice of the optimal step size is actually mostly influenced by the values of $\mu$, $L$ and $\gamma$. These (hyper)parameters are central and in order to show this, we study two key quantities, namely the spectral radius of the iteration matrix and the covariance matrix associated with the NAG-GS method summarized by Algorithm 1. Note that this theoretical study only concerns the case $f(x) = \frac{1}{2}x^T A x$. Considering the size limitation of the paper, we present below only the main theoretical result and place its proof in supplementary materials, Section 1.1.4.:

**Theorem 1.** *For $G_{NAG}$ equation 7, given $\gamma \geq \mu$, and assuming $0 < \mu = \lambda_1 \leq \ldots \leq \lambda_n = L < \infty$; if $0 < \alpha \leq \frac{\mu + \gamma + \sqrt{(\mu - \gamma)^2 + 4\gamma L}}{L - \mu}$, then the NAG-GS method summarized by Algorithm 1 is convergent for the $n$-dimensional case, with $n > 2$.*

**Remark 2.** *It is important to mention that the optimal contraction rate of NAG-GS aiming at minimizing a strongly convex quadratic function is reached for $\alpha = \frac{\mu + \gamma + \sqrt{(\mu - \gamma)^2 + 4\gamma L}}{L - \mu}$.*

All the steps of the convergence analysis are fully detailed in supplementary materials, Section 1.1, and organized as follows:

- Sections 1.1.1. and 1.1.2. in supplementary materials respectively provide the full analysis of the spectral radius of the iteration matrix associated with the NAG-GS method and the covariance matrix at stationarity w.r.t. hyperparameters $\mu$, $L$, $\gamma$ and $\sigma$, for the case of the dimension $n = 2$. The theoretical results obtained are summarized in Section 1.1.3 in supplementary materials to come up with an optimal step size in terms of contraction rate. The extension to $n > 2$ is detailed in Section 1.1.4 along with the proof of Theorem 1.

- Numerical tests are performed and detailed in supplementary materials, Section 1.1.5, to support the theoretical results obtained for the quadratic case.

## 3 EXPERIMENTS

We test the NAG-GS method on several neural architectures: logistic regression, transformer model for natural language processing (RoBERTa model) and computer vision (ViT model) tasks, residual networks for computer vision tasks (ResNet20). To ensure a fair benchmark of our method on these neural architectures, we replace the reference optimizers with our own and solely adjust the hyperparameters of our optimizer. We maintain the integrity of the model architectures and hyperparameters, including the dropout rate, schedule, batch size, number of training epochs, and evaluation methodology. The experiments described below can be easily reproduced using the available codes[1]. The results of the benchmark for the considered models are summarized in Table 1.

Table 1: Summary on the comparison of NAG-GS to the reference optimizer for different neural architectures (greater is better). Target metrics are ACC@1 for RESNET20 and VIT, and the average score on GLUE for ROBERTA.

| MODEL | DATASET | OPTIMIZER | SCORE |
|---|---|---|---|
| ResNet20 | CIFAR-10 | SGD-MW | 91.25 |
| | | NAG-GS | **91.29** |
| RoBERTa | GLUE | AdamW | **82.92** |
| | | NAG-GS | 82.44 |
| ViT | food101 | AdamW | 83.24 |
| | | NAG-GS | **86.06** |

### 3.1 TOY PROBLEMS

In this section, we illustrate the convergence of the NAG-GS method for a strongly convex quadratic function and a one-dimensional non-convex function. These experiments demonstrate that the interval of the feasible learning rates for NAG-GS is larger than for competitors.

**Strongly convex quadratic function.** Consider the problem $\min_x f(x)$, where $f(x) = \frac{1}{2}x^\top A x - b^\top x$ is convex quadratic function. The matrix $A \in \mathbb{S}_{++}^n$ is symmetric and positive semidefinite, $L = \lambda_{\max}(A)$, $\mu = \lambda_{\min}(A)$ and $n = 100$. Figure 1 shows the dependence of the number of iterations needed for convergence of NAG-GS, gradient descent (GD), accelerated gradient descent (AGD) and Heavy ball method (HB) on the learning rates for different $\mu$ and $L$. A method converges if $f(x_k) - f^* \le 10^{-4}$, where $f^* = f(x^*)$ is the optimum function value. If the learning rate leads to divergence, we set the number of iterations to $10^{10}$. Figure 1 shows that NAG-GS provides two benefits. First, it accepts larger learning rates compared to GD, AGD, and HB methods. Second, NAG-GS converges faster in terms of the number of iterations compared to GD, AGD, and HB methods in the large learning rate regime. In this experiment, we use the version of accelerated gradient descent from Su et al. (2014). In NAG-GS we use constant $\gamma = \mu = \lambda_{\min}(A)$. In HB, we use constant $\beta = 0.9$. Also, we test 70 learning rates distributed uniformly in the logarithmic grid in the interval $[10^{-3}, 10]$.

### 3.2 LOGISTIC REGRESSION

In this section, we benchmark NAG-GS method against state-of-the-art optimizers on the logistic regression training problem for MNIST dataset LeCun et al. (2010). Since this problem is convex and non-quadratic, we consider this problem as the natural and next test case after the theoretical analysis and numerical tests of the NAG-GS method in Section 2.3 for the quadratic convex problem. In Figure 2 and Table 2 we present the comparison of the NAG-GS method with competitors. We confirm numerically that the NAG-GS method allows the use of a larger range of values for the learning rate than SGD Momentum and AdamW optimizers. This observation highlights the robustness of our method w.r.t. the selection of hyperparameters. Moreover, the results indicate that the semi-implicit nature of the NAG-GS method indeed ensures the acceleration effect through the

---

[1]`https://github.com/naggsopt/naggs`

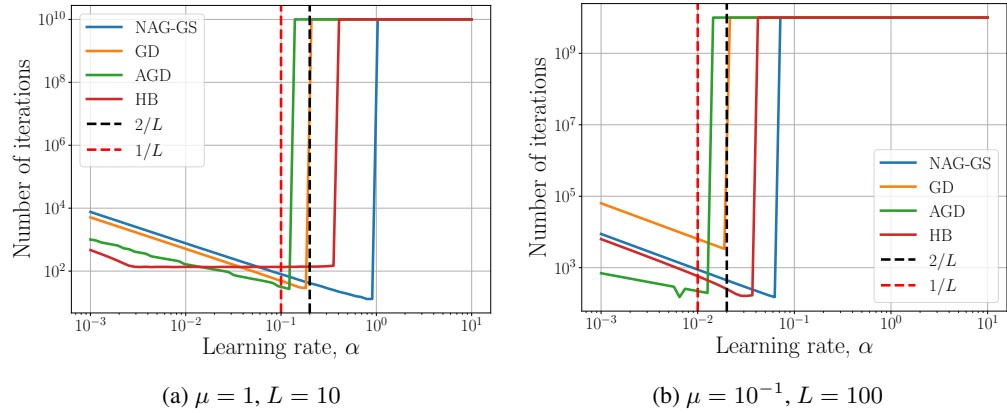

(a) $\mu = 1$, $L = 10$        (b) $\mu = 10^{-1}$, $L = 100$

Figure 1: Dependence of the number of iterations needed for convergence on the learning rate used in the corresponding method. NAG-GS is more robust with respect to the learning rate than gradient descent (GD), accelerated gradient descent (AGD) and Heavy ball method (HB). Also, NAG-GS converges faster than competitors if the learning rate is sufficiently large. The number of iterations $10^{10}$ indicates the divergence of the method with a corresponding learning rate.

use of larger learning rates while keeping a high accuracy of the model, and this holds not only for the convex quadratic problems but also for non-quadratic convex ones.

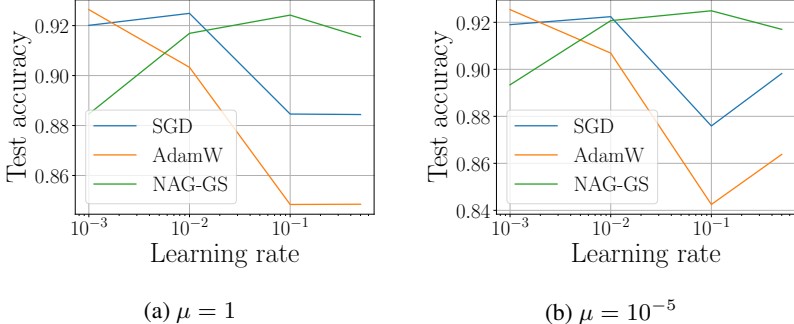

(a) $\mu = 1$        (b) $\mu = 10^{-5}$

Figure 2: Dependence of the test accuracy on the learning rates for the considered methods. NAG-GS provides the highest test accuracy for the larger learning rate. This trend preserves for considered $\mu$ of different orders.

Table 2: Test accuracies for NAG-GS, SGD-Momentum, and AdamW for the logistic regression model and MNIST classification problem. NAG-GS gives higher test accuracy for large learning rates, which indicates that it is more robust and does not diverge while learning rate is increased.

| Learning rate | NAG-GS | SGD | AdamW |
|---|---|---|---|
| $10^{-3}$ | 0.8934 | 0.9190 | **0.9254** |
| $10^{-2}$ | 0.9207 | **0.9224** | 0.9069 |
| 0.1 | **0.9249** | 0.8759 | 0.8425 |
| 0.5 | **0.9170** | 0.8982 | 0.8638 |

## 3.3 TRANSFORMER MODELS

### 3.3.1 ROBERTA

In this section we test NAG-GS optimizer in the frame of natural language processing for the tasks of fine-tuning pretrained model on GLUE benchmark datasets Wang et al. (2018). We use pretrained RoBERTa Liu et al. (2019) model from Hugging Face's TRANSFORMERS Wolf et al. (2020) library.

In this benchmark, the reference optimizer is AdamW Ilya et al. (2019) with polynomial learning rate schedule. The training setup defined in Liu et al. (2019) is used for both NAG-GS and AdamW optimizers. We search for an optimal learning rate for NAG-GS optimizer with fixed $\gamma$ and $\mu$ to get the best performance on the task at hand. Note that NAG-GS is used with constant schedule which makes it simpler to tune. In terms of learning rate values, the one allowed by AdamW is around $10^{-5}$ while NAG-GS allows a much bigger value of $10^{-2}$. Evaluation results on GLUE tasks are presented in Table 3. Despite a rather restrained search space for NAG-GS hyperparameters, it demonstrates better performance on some tasks and competitive performance on others. Figure 3 shows the behavior of loss values and target metrics on GLUE.

Table 3: Comparison of AdamW and NAG-GS optimizers in fine-tuning on GLUE benchmark. We use reported hyperparameters for AdamW. In the case of NAG-GS, we search hyperparameters space for the best performance metric. Search space consists of learning rate $\alpha$ from $[10^{-3}, 10^{0}]$, factor $\gamma$ from $[10^{-2}, 10^{0}]$, and momentum $\mu = 1$.

| OPTIMIZER | CoLA | MNLI | MRPC | QNLI | QQP | RTE | SST2 | STS-B | WNLI |
|---|---|---|---|---|---|---|---|---|---|
| ADAMW | **61.60** | **87.56** | 88.24 | **92.62** | **91.69** | **78.34** | **94.95** | 90.68 | **56.34** |
| NAG-GS | **61.60** | 87.24 | **90.69** | 92.59 | 91.01 | 77.97 | 94.50 | 90.21 | **56.34** |

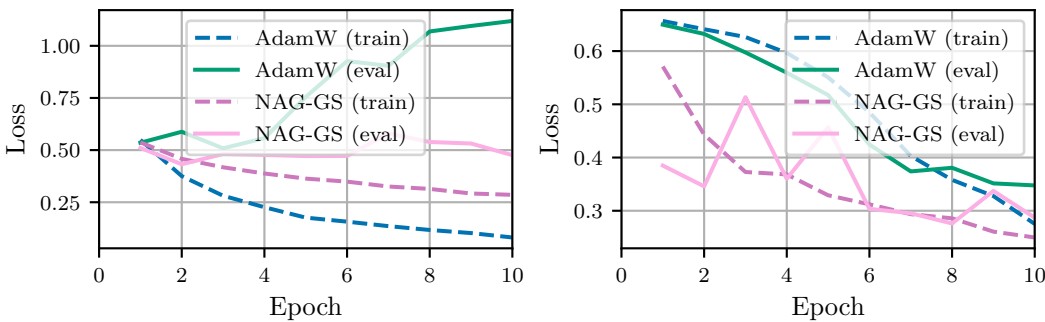

Figure 3: Cross-entropy losses on validation and train sets for CoLA (left) and MRPC (right) tasks. Solid lines correspond to the best trial with the NAG-GS optimizer.

### 3.3.2 VISION TRANSFORMER MODEL

We used the Vision Transformer model Wu et al. (2020), which was pretrained on the ImageNet dataset Deng et al. (2009), and fine-tuned it on the `food101` dataset Bossard et al. (2014) using NAG-GS and AdamW. It is worth noting that all weights were updated during the fine-tuning. This task involves classifying a dataset of 101 food categories, with 1000 images per class. To ensure a fair comparison, we first conducted an intensive hyperparameter search Biewald (2020) for all possible hyperparameter configurations on a subset of the data for each of the methods and selected the best configuration. After the hyperparameter search, we performed the experiments on the entire dataset. The results are presented in Table 4. We observed that properly-tuned NAG-GS outperformed AdamW in both training and evaluation metrics. Also, NAG-GS reached higher accuracy compared to AdamW after one epoch. The optimal hyperparameters found for NAG-GS are $\alpha = 0.07929, \gamma = 0.3554, \mu = 0.1301$; for AdamW lr $= 0.00004949, \beta_1 = 0.8679, \beta_2 = 0.9969$.

Table 4: Test accuracies for NAG-GS and AdamW.

| Stage | NAG-GS | AdamW |
|---|---|---|
| After 1 epoch | **0.8419** | 0.8269 |
| After 25 epochs | **0.8606** | 0.8324 |

## 3.4 ResNet-20 and VGG-11

We compare NAG-GS and momentum SGD with weight decay (SGD-MW) on ResNet-20 He et al. (2016) and VGG-11 Simonyan & Zisserman (2014) models. In particular, we choose these architectures for versatile experimental verification of properties of our optimizer.

**ResNet-20.**    We carried out intensive experiments in order to deeply evaluate the performance of NAG-GS for computer vision tasks (residual networks in particular) and to show that NAG-GS with the appropriate choice of optimizer parameters is on par with SGD-MW (see Table 1 and Figure 4). For the latter, we use the parameters reported in the literature. The classification problem is solved using CIFAR-10 Krizhevsky (2009). The experimental setup is the same in all experiments except optimizer and its parameters. The best test score for NAG-GS is achieved for $\alpha = 0.11$, $\gamma = 17$, and $\mu = 0.01$.

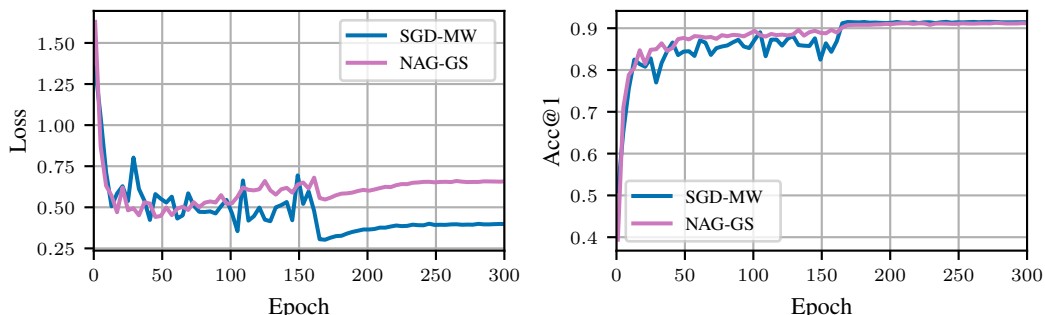

Figure 4: Evaluation of NAG-GS with SGD-MW on ResNet-20 on CIFAR-10.

**VGG-11.**    We test this architecture on the CIFAR-10 image classification problem without data resizing and demonstrate the robustness of the NAG-GS optimizer to large learning rates compared to SGD-MW. The hyperparameters are the following: batch size equals to 1000, number of epoch is 50. We use the constant $\gamma = 1$. and $\mu = 10^{-4}$ equal to the weight decay parameter in SGD-MW. Also, momentum term in SGD-MW equals to $0.9$. Comparison results are presented in Table 5, where the resulting test accuracy after 50 epochs are given. From this table follows that NAG-GS preserves the expected behaviour to show higher test accuracy in the large learning rate regime compared to SGD-MW optimizer.

Table 5: Test accuracies for NAG-GS and SGD-MW (SGD with momentum and weight decay) for CIFAR-10 classification task on VGG-11 model. NAG-GS gives higher test accuracy for large learning rates to confirm that it is more robust and does not diverge while learning rate is increased.

| Learning rate | NAG-GS | SGD-MW |
|---|---|---|
| $10^{-3}$ | 0.1 | **0.65** |
| $10^{-2}$ | 0.62 | **0.74** |
| 0.1 | **0.76** | 0.1 |
| 0.2 | **0.76** | 0.1 |

## 4 Related works

The approach of interpreting and analyzing optimization methods from the ODEs discretization perspective is well-known and widely used in practice (Muehlebach & Jordan, 2019; Wilson et al., 2021; Shi et al., 2021; Alvarez & Attouch, 2001; Merkulov & Oseledets, 2020). The main advantage of this approach is to construct a direct correspondence between the properties of some classes of ODEs and their associated optimization methods. In particular, gradient descent and Nesterov accelerated methods are discussed in (Su et al., 2014) as a particular discretization of ODEs. In the same perspective, many other optimization methods were analyzed, we can mention the mirror descent method and its accelerated versions (Krichene et al., 2015), the proximal methods (Attouch

et al., 2019) and ADMM (Franca et al., 2018). It is well known that discretization strategy is essential for transforming a particular ODE to an efficient optimization method, Shi et al. (2019); Zhang et al. (2018) investigate the most proper discretization techniques for different classes of ODEs. A similar analysis but for stochastic first-order methods is presented in (Laborde & Oberman, 2020; Malladi et al., 2022). Recent advances in deriving optimal optimizers (Taylor & Drori, 2023; Zhou et al., 2020) do not exploit the ODE interpretations, which is an interesting future work, and do not consider stochastic setup.

## 5 CONCLUSIONS AND FURTHER WORKS

We have presented a new and theoretically motivated stochastic optimizer called NAG-GS. It comes from the semi-implicit Gauss-Seidel type discretization of a well-chosen accelerated Nesterov-like SDE. These building blocks ensure two central properties for NAG-GS: (1) the ability to accelerate the optimization process and (2) better robustness to large learning rates. We demonstrate these features theoretically and provide a detailed analysis of the convergence of the method in the quadratic case. Moreover, we show that NAG-GS is competitive with state-of-the-art methods for tackling a wide variety of stochastic optimization problems of increasing complexity and dimension, starting from the logistic regression model to the training of large machine learning models such as ResNet-20, VGG-11 and Transformers. In all tests, NAG-GS demonstrates competitive performance compared with standard optimizers. Further works will focus on the non-asymptotic convergence analysis of NAG-GS for general convex functions, the derivation of efficient and tractable higher-order methods based on the full-implicit discretization of the accelerated Nesterov-like SDE, and the introduction of variants of NAG-GS tailored for gradient noise with unbounded variance.

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
