# SUPPLEMENTARY MATERIALS

**Anonymous authors**

## 1  ADDITIONAL REMARKS RELATED TO THEORETICAL BACKGROUND

An accelerated ODE has been presented in the main text Section 1.1 which relied on a specific spectral transformation. In this brief section, we add some useful insights:

- Equation (10) is a variant of the heavy ball model with variable damping coefficients in front of $\ddot{x}$ and $\dot{x}$.

- Thanks to the scaling factor $\gamma$, both the convex case $\mu = 0$ and the strongly convex case $\mu > 0$ can be handled in a unified way.

- In the continuous time, one can solve easily (8) as follows: $\gamma(t) = \mu + (\gamma_0 - \mu)e^{-t}, \quad t \geq 0$. Since $\gamma_0 > 0$, we have that $\gamma(t) > 0$ for all $t \geq 0$ and $\gamma(t)$ converges to $\mu$ exponentially and monotonically as $t \to +\infty$. In particular, if $\gamma_0 = \mu > 0$, then $\gamma(t) = \mu$ for all $t \geq 0$. We remark here the links between the behavior of the scaling factor $\gamma(t)$ and the sequence $\{\gamma_k\}_{k=0}^{\infty}$ introduced by Nesterov Nesterov (2018) in its analysis of optimal first-order methods in discrete-time, see (Nesterov, 2018, Lemma 2.2.3).

- Authors from Luo & Chen (2021) prove the exponential decay property $\mathcal{L}(t) \leq e^{-t}\mathcal{L}_0, \quad t > 0$ for a Taylored Lyapunov function $\mathcal{L}(t) := f(x(t)) - f(x^{\star}) + \frac{\gamma(t)}{2}\|v(t) - x^{\star}\|^2$ where $x^{\star} \in \text{argmin } f$ is a global minimizer of $f$. Again we note the similarity between the Lyapunov function proposed here and the estimating sequence $\{\phi_k(x)\}_{k=0}^{\infty}$ of function $f$ introduced by Nesterov in its optimal first-order methods analysis Nesterov (2018). In (Nesterov, 2018, Lemma 2.2.3), this sequence that takes the form $\phi_k(x) = \phi_k^{\star}(x) + \frac{\gamma_k}{2}\|v_k - x\|^2$ where $\gamma_{k+1} := (1 - \alpha_k)\gamma_k + \alpha_k\mu$ and $v_{k+1} := \frac{1}{\gamma_{k+1}}[(1 - \alpha_k)\gamma_k v_k + \alpha_k\mu y_k - \alpha_k\nabla f(y_k)]$ which stand for a forward Euler discretization respectively of (8) and second ODE of (9).

We ask the attentive reader to remember that this discussion mainly concerns the continuous time case. A second central part of our analysis was based on the methods of discretization of (9). Indeed, these discretizations ensure together with the spectral transformation (7) the optimal convergence rates of the methods and their particular ability to handle noisy gradients.

Finally, we delve into a crucial insight motivating the proposition of an optimizer that exhibits robustness concerning the choice of the step size, enabling the utilization of a wide range of values for the step size (or learning rate). An established approach for analyzing Stochastic Gradient Descent (SGD) involves viewing it as a discretization of a continuous-time process, expressed as:

$$\mathrm{d}x_t = -\nabla f(x_t)\mathrm{d}t + \sqrt{\alpha\sigma^2}\mathrm{d}B_t,$$

where $B_t$ denotes the standard Brownian motion. This stochastic differential equation (SDE) is a variant of the well-known Langevin diffusion. Under mild regularity assumptions on $f$, it can be shown that the Markov process $(x_t)_{t \geq 0}$ is ergodic, with its unique invariant measure having a density proportional to $\exp\left(-f(x)/(\alpha\sigma^2)\right)$ for any $\alpha > 0$ Roberts & Stramer (2002).

Building on this observation, existing research has shed light on the relationship between the invariant measure and algorithm parameters. Jastrzębski et al. (2018) focused on the interplay of the invariant measure with step-size ($\alpha$) and mini-batch size, as a function of $\sigma^2$. They concluded that the ratio of learning rate to batch size serves as the control parameter determining the width of the minima found by SGD.

Additionally, Keskar et al. (2017) explored sharp and flat minimizers, their impact on generalization, and the distinctions between large-batch and small-batch methods, especially in the context of deep neural networks. Their key observations include:

- **Sharp and Flat Minimizers:** Flat minimizers exhibit slow function variation in a wide neighborhood, while sharp minimizers show rapid increases in a small neighborhood. Flat minimizers can be described with lower precision, contrasting with the higher precision needed for sharp minimizers.

- **Effect on Generalization:** Sharp minimizers negatively affect model generalization due to their large sensitivity in the training function. The Minimum Description Length (MDL) theory suggests that lower complexity models generalize better, making flat minimizers preferable.

- **Large-batch vs. Small-batch Methods:** Large-batch methods tend to converge to sharp minimizers, resulting in reduced generalization ability. Conversely, small-batch methods converge to flat minimizers, generally leading to better generalization.

- **Observation in Deep Neural Networks:** The loss function landscape in deep neural networks attracts large-batch methods towards regions with sharp minimizers, trapping them and impeding their escape from these basins of attraction.

Motivated by these insights and assuming a Gaussian noise process, it becomes evident that proposing a solver capable of handling a broad range of step size values is crucial. Specifically, allowing for a larger ratio $\alpha/b$ (with $b$ as the mini-batch size) increases the likelihood of converging to wider local minima, ultimately enhancing the generalization performance of the trained model.

## 1.1 CONVERGENCE/STABILITY ANALYSIS OF THE QUADRATIC CASE: DETAILS

As briefly mentioned in Section 2.3 of the main text, the two key elements to come up with a maximum (constant) step size for Algorithm 1 are the study of the spectral radius of iteration matrix associated with NAG-GS scheme (Section 1.1.1) and the covariance matrix at stationarity (Section 1.1.2) w.r.t. all the significant parameters of the scheme. These parameters are the step size (integration step/time step) $\alpha$, the convexity parameters $0 \leq \mu \leq L \leq \infty$ of the function $f(x)$, the variance of the noise $\sigma^2$ and the positive scaling parameter $\gamma$. Note that this theoretical study only concerns the case $f(x) = \frac{1}{2}x^\top A x$.

**Reproducibility**

- In Section 1.1.1, we start by determining the explicit formulation of the spectral radius of the iteration matrix $\rho(E(\alpha))$, specifically for the 2-dimensional quadratic case. This formulation allows us to derive the optimal step size $\alpha_c$ that minimizes $\rho(E(\alpha))$, resulting in the highest convergence rate for NAG-GS method. Notably, Lemma 2 presents a crucial outcome for the asymptotic convergence analysis of NAG-GS, revealing that $\rho(E(\alpha))$ is a strictly monotonically increasing function of $\alpha$ within a certain interval, under mild assumptions.

- In Section 1.1.2, we conduct an in-depth analysis of the covariance matrix at stationarity, which enables us to establish the sufficient conditions for $\alpha_c$ to ensure the asymptotic convergence of the NAG-GS method. The formal proof for this convergence is presented in Lemma 3 for the case of $n = 2$.

- In Section 1.1.4, we provide the formal proof of Theorem 1, which is enunciated in the main text. This theorem stated the asymptotic convergence of the NAG-GS method for dimensions $n > 2$.

### 1.1.1 SPECTRAL RADIUS ANALYSIS

Let us assume $f(x) = \frac{1}{2}x^\top A x$ and since $A \in \mathbb{S}_+^n$ by hypothesis, it is diagonalizable and can be presented as $A = \text{diag}(\lambda_1, \ldots, \lambda_n)$ without loss of generality, that is to say, that we will consider a system of coordinates composed of the eigenvectors of matrix $A$. Let us note that $\mu = \lambda_1 \leq \ldots \leq \lambda_n = L$.

For the following we restrict the discussion to the case $n = 2$. In this setting, $y = (x, v) \in \mathbb{R}^4$ and the matrices $M$ and $N$ from the Gauss-Seidel splitting of $G_{NAG}$ (7) are:

$$M = \begin{bmatrix} -I_{2\times2} & 0_{2\times2} \\ \mu/\gamma I_{2\times2} - A/\gamma & -\mu/\gamma I_{2\times2} \end{bmatrix} = \begin{bmatrix} -1 & 0 & 0 & 0 \\ 0 & -1 & 0 & 0 \\ 0 & 0 & -\mu/\gamma & 0 \\ 0 & \mu/\gamma - L/\gamma & 0 & -\mu/\gamma \end{bmatrix},$$

$$N = \begin{bmatrix} 0_{2\times2} & I_{2\times2} \\ 0_{2\times2} & 0_{2\times2} \end{bmatrix}$$

For the minimization of $f(x) = \frac{1}{2}x^\top A x$, given the property of Brownian motion $\Delta W_k = W_{k+1} - W_k = \sqrt{\alpha_k}\eta_k$ where $\eta_k \sim \mathcal{N}(0,1)$, (12) reads:

$$y_{k+1} = (I_{4\times4} - \alpha M)^{-1}(I_{4\times4} + \alpha N)y_k + (I_{4\times4} - \alpha M)^{-1}\begin{bmatrix} 0 \\ \sigma\sqrt{\alpha}\eta_k \end{bmatrix} \tag{1}$$

Since matrix $M$ is lower-triangular, matrix $I_{4\times4} - \alpha M$ is as well and can be factorized as follows:

$$I_{4\times4} - \alpha M = DT$$
$$= \begin{bmatrix} (1+\alpha)I_{2\times2} & 0_{2\times2} \\ 0_{2\times2} & (1+\frac{\alpha\mu}{\gamma})I_{2\times2} \end{bmatrix} \begin{bmatrix} I_{2\times2} & 0_{2\times2} \\ \frac{\alpha(A-\mu I_{2\times2})}{\gamma(1+\frac{\alpha\mu}{\gamma})} & I_{2\times2} \end{bmatrix}$$

Hence $(I_{4\times4} - \alpha M)^{-1} = T^{-1}D^{-1}$ where $D^{-1}$ can be easily computed. It remains to compute $T^{-1}$; $T$ can be decomposed as follows: $T = I_{4\times4} + Q$ with $Q$ a nilpotent matrix such that $QQ = O_{4\times4}$. For such decomposition, it is well known that:

$$T^{-1} = (I_{4\times4} + Q)^{-1} = I_{4\times4} - Q = \begin{bmatrix} I_{2\times2} & 0_{2\times2} \\ \frac{\alpha(\mu I_{2\times2}-A)}{\gamma(1+\tau_k)} & I_{2\times2} \end{bmatrix} \tag{2}$$

where $\tau_k = \frac{\alpha\mu}{\gamma}$. Combining these results, (1) finally reads:

$$y_{k+1} = \begin{bmatrix} \frac{1}{\alpha+1} & 0 & \frac{\alpha}{1+\alpha} & 0 \\ 0 & \frac{1}{\alpha+1} & 0 & \frac{\alpha}{1+\alpha} \\ 0 & 0 & \frac{1}{1+\tau} & 0 \\ 0 & \frac{\alpha(\mu-L)}{\gamma(\tau+1)(\alpha+1)} & 0 & \frac{\alpha^2(\mu-L)}{\gamma(1+\tau)(1+\alpha)} + \frac{1}{1+\tau} \end{bmatrix} y_k + \begin{bmatrix} 0 \\ \sigma\frac{\sqrt{\alpha}}{1+\tau}\eta_k \end{bmatrix}$$
$$= Ey_k + \begin{bmatrix} 0 \\ \sigma\frac{\sqrt{\alpha}}{1+\tau}\eta_k \end{bmatrix} \tag{3}$$

with $E$ denoting the iteration matrix associated with the NAG-GS method. Hence (3) includes two terms, the first is the product of the iteration matrix times the current vector $y_k$ and the second one features the effect of the noise. For the latter, it will be studied in Section 1.1.2 from the point of view of maximum step size for the NAG-GS method through the key quantity of the covariance matrix. Let us focus on the first term. It is clear that in order to get the maximum contraction rate, we should look for $\alpha$ that minimizes the spectral radius of $E$. Since the spectral radius is the maximum absolute value of the eigenvalues of iteration matrix $E$, we start by computing them. Let us find the expression of $\lambda_i \in \sigma(E)$ for $1 \le i \le 4$ that satisfies $\det(E - \lambda I_{4\times4}) = 0$ as functions of the scheme's parameters. Solving

$$\det(E - \lambda I_{4\times4}) = 0$$
$$\equiv \frac{(\gamma\lambda - \gamma + \alpha\lambda\mu)(\lambda + \alpha\lambda - 1)(\gamma - 2\gamma\lambda + \gamma\lambda^2 + \alpha^2\lambda^2\mu - \alpha\gamma\lambda - \alpha\lambda\mu + L\alpha^2\lambda + \alpha\gamma\lambda^2 + \alpha\lambda^2\mu - \alpha^2\lambda\mu)}{(\alpha+1)^2(\gamma+\alpha\mu)^2} = 0 \tag{4}$$

leads to the following eigenvalues:

$$\lambda_1 = \frac{\gamma}{\gamma + \alpha\mu}$$

$$\lambda_2 = \frac{1}{1 + \alpha}$$

$$\lambda_3 = \frac{2\gamma + \alpha\gamma + \alpha\mu - L\alpha^2 + \alpha^2\mu}{2(\gamma + \alpha\gamma + \alpha\mu + \alpha^2\mu)} +$$

$$\frac{\alpha\sqrt{L^2\alpha^2 - 2L\alpha^2\mu - 2L\alpha\mu - 2\gamma L\alpha - 4\gamma L + \alpha^2\mu^2 + 2\alpha\mu^2 + 2\gamma\alpha\mu + \mu^2 + 2\gamma\mu + \gamma^2}}{2(\gamma + \alpha\gamma + \alpha\mu + \alpha^2\mu)} \quad (5)$$

$$\lambda_4 = \frac{2\gamma + \alpha\gamma + \alpha\mu - L\alpha^2 + \alpha^2\mu}{2(\gamma + \alpha\gamma + \alpha\mu + \alpha^2\mu)} -$$

$$\frac{\alpha\sqrt{L^2\alpha^2 - 2L\alpha^2\mu - 2L\alpha\mu - 2\gamma L\alpha - 4\gamma L + \alpha^2\mu^2 + 2\alpha\mu^2 + 2\gamma\alpha\mu + \mu^2 + 2\gamma\mu + \gamma^2}}{2(\gamma + \alpha\gamma + \alpha\mu + \alpha^2\mu)}$$

Let us first mention some general behavior or these eigenvalues. Given $\gamma$ and $\mu$ positive, we observe that:

1. $\lambda_1$ and $\lambda_2$ are positive decreasing functions w.r.t. $\alpha$. Moreover, for bounded $\gamma$ and $\mu$, we have $\lim_{\alpha\to\infty} |\lambda_1(\alpha)| = 0 = \lim_{\alpha\to\infty} |\lambda_2(\alpha)|$.

2. One can show that for $\alpha \in [\frac{\mu+\gamma-2\sqrt{\gamma L}}{L-\mu}, \frac{\mu+\gamma+2\sqrt{\gamma L}}{L-\mu}]$, functions $\lambda_3(\alpha)$ and $\lambda_4(\alpha)$ are complex values and one can easily show that both share the same absolute value. Note that the lower bound of the interval $\frac{\mu+\gamma-2\sqrt{\gamma L}}{L-\mu}$ is negative as soon as $\gamma \in [2L - \mu - 2\sqrt{L^2 - \mu L}, 2L - \mu + 2\sqrt{L^2 - \mu L}] \subseteq \mathbb{R}_+$. Moreover, one can easily show that $\lim_{\alpha\to\infty} |\lambda_3(\alpha)| = 0$ and $\lim_{\alpha\to\infty} |\lambda_4(\alpha)| = \frac{L-\mu}{\mu} = \kappa(A) - 1$. The latter limit shows that eigenvalue $\lambda_4$ plays a central role in the convergence of the NAG-GS method since it is the one that can reach the value one and violate the convergence condition, as soon as $\kappa(A) > 2$. The analysis of $\lambda_4$ also allows us to come up with a good candidate for the step size $\alpha$ that minimizes the spectral radius of matrix $E$, especially and obviously at critical point $\alpha_{max} = \frac{\mu+\gamma+2\sqrt{\gamma L}}{L-\mu}$ which is positive since $L \geq \mu$ by hypothesis. Note that the case $L \to \mu$ gives some preliminary hints that the maximum step size can be almost "unbounded" in some particular cases.

Now, let us study these eigenvalues in more detail, it seems that three different scenarios must be studied:

1. For any variant of Algorithm 1 for which $\gamma_0 = \mu$, then $\gamma = \mu$ for all $k \geq 0$ and therefore $\lambda_1(\alpha) = \lambda_2(\alpha)$. Moreover, at $\alpha = \frac{\mu+\gamma+2\sqrt{\gamma L}}{L-\mu} = \frac{2\mu+2\sqrt{\mu L}}{L-\mu}$, we can easily check that $|\lambda_1(\alpha)| = |\lambda_2(\alpha)| = |\lambda_3(\alpha)| = |\lambda_4(\alpha)|$. Therefore $\alpha = \frac{2\mu+2\sqrt{\mu L}}{L-\mu}$ is the step size ensuring the minimal spectral radius and hence the maximum contraction rate. Figure 1 shows the evolution of the absolute values of the eigenvalues of iteration matrix $E$ w.r.t. $\alpha$ for such a setting.

2. As soon as $\gamma < \mu$, one can easily show that $\lambda_1(\alpha) < \lambda_2(\alpha)$. Therefore the step size $\alpha$ with the minimal spectral radius is such that $|\lambda_4(\alpha)| = |\lambda_2(\alpha)|$. One can show that the equality holds for $\alpha = \frac{\mu+\gamma+\sqrt{(\mu-\gamma)^2+4\gamma L}}{L-\mu}$. One can easily check that $\frac{\mu+\gamma+\sqrt{(\mu-\gamma)^2+4\gamma L}}{L-\mu} - \frac{\mu+\gamma+2\sqrt{\gamma L}}{L-\mu} = (\mu-\gamma)^2 > 0$. Hence the second candidate for step size $\alpha$ will be bigger than the first one and the distance between them increases as the squared distance between $\gamma$ and $\mu$. Figure 2 shows the evolution of the absolute values of the eigenvalues of iteration matrix $E$ w.r.t. $\alpha$ for this setting.

3. For $\gamma > \mu$: the analysis of this case gives the same results as the previous point. According to Algorithm 1, $\gamma$ is either constant and equal to $\mu$ or decreasing to $\mu$ along iterations. Hence, the case $\gamma > \mu$ will be considered for the theoretical analysis when $\gamma \neq \mu$.

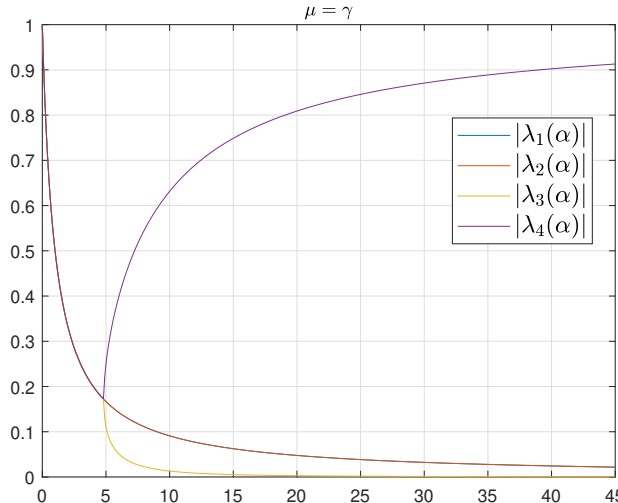

Figure 1: Evolution of absolute values of $\lambda_i$ w.r.t $\alpha$; $\mu = \gamma$.

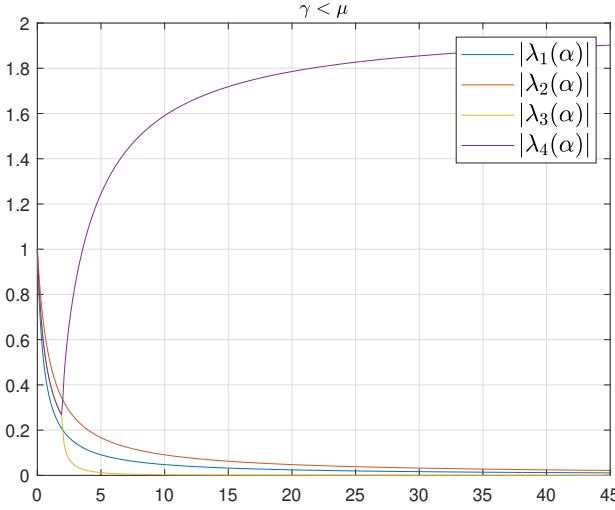

Figure 2: Evolution of absolute values of $\lambda_i$ w.r.t $\alpha$; $\gamma < \mu$.

As a first summary, the detailed analysis of the eigenvalues of iteration matrix $E$ w.r.t. the significant parameters of the NAG-GS method leads us to come up with two candidates for the step size that minimize the spectral radius of $E$, hence ensuring the highest contraction rate possible. These results will be gathered with those obtained in Section 1.1.2 dedicated to the covariance matrix analysis.

Let us now look at the behavior of the dynamics in expectation; given the properties of the Brownian motion and by applying the Expectation operator $\mathbb{E}$ on both sides of the system of SDE's (11), the resulting "averaged" equations identify with the "deterministic" setting studied by Luo & Chen (2021). For such a setting, authors from Luo & Chen (2021) demonstrated that, if $0 \le \alpha \le \frac{2}{\sqrt{\kappa(A)}}$, then a Gauss–Seidel splitting-based scheme for solving (9) is A-stable for quadratic objectives in the deterministic setting. We conclude this section by showing that the two candidates we derived above for step size are higher than the limit $\frac{2}{\sqrt{\kappa(A)}}$ given in (Luo & Chen, 2021, Theorem 1). It can be intuitively understood in the case $L \to \mu$, however, we give a formal proof in Lemma 1.

**Lemma 1.** *Given $\gamma > 0$, and assuming $0 < \mu < L$, then for $\gamma = \mu$ and $\gamma > \mu$ the following inequalities respectively hold:*

$$\frac{2\mu + 2\sqrt{\mu L}}{L - \mu} > \frac{2}{\sqrt{\kappa(A)}}$$

$$\frac{\mu + \gamma + \sqrt{(\mu - \gamma)^2 + 4\gamma L}}{L - \mu} > \frac{2}{\sqrt{\kappa(A)}} \tag{6}$$

*where $\kappa(A) = \frac{L}{\mu}$.*

*Proof.* Let us start for the case $\mu = \gamma$, hence first inequality from (6) becomes:

$$\frac{2\mu + 2\sqrt{L\mu}}{L - \mu} > \frac{2}{\sqrt{L/\mu}}$$

$$\equiv (\mu + \sqrt{L\mu})\sqrt{L/\mu} > (L - \mu)$$

$$\equiv \sqrt{\mu L} + L > L - \mu$$

$$\equiv \sqrt{\mu L} > -\mu$$

which holds for any positive $\mu, L$ and satisfied by hypothesis. For the case $\gamma > \mu$, we have:

$$\frac{\mu + \gamma + \sqrt{(\mu - \gamma)^2 + 4\gamma L}}{L - \mu} > \frac{2}{\sqrt{L/\mu}}$$

$$\equiv \sqrt{(\mu - \gamma)^2 + 4\gamma L} > \frac{2}{\sqrt{L/\mu}}(L - \mu) - \gamma - \mu$$

$$\equiv (\mu - \gamma)^2 + 4\gamma L > (\mu + 2\sqrt{\frac{\mu}{L}}(\mu - L) + \gamma)^2$$

$$\equiv \gamma > \frac{-2\mu^2 + \mu^3/L + \mu^2\sqrt{\mu/L} + \mu L - \mu L\sqrt{\mu/L}}{-\mu - \sqrt{\mu/L}(\mu - L) + L}$$

where second inequality hold since $L \geq \mu$ and last inequality holds since $-\mu - \sqrt{\mu/L}(\mu - L) + L > 0$ (one can easily check this by using $L > \mu$). It remains to show that:

$$\mu > \frac{-2\mu^2 + \mu^3/L + \mu^2\sqrt{\mu/L} + \mu L - \mu L\sqrt{\mu/L}}{-\mu - \sqrt{\mu/L}(\mu - L) + L}$$

which holds for any $\mu$ and $L$ positive (technical details are skipped; it mainly consists of the study of a table of signs of a polynomial equation in $\mu$).

Since $\gamma > \mu$ by hypothesis, therefore inequality

$$\gamma > \frac{-2\mu^2 + \mu^3/L + \mu^2\sqrt{\mu/L} + \mu L - \mu L\sqrt{\mu/L}}{-\mu - \sqrt{\mu/L}(\mu - L) + L}$$

holds for any $\mu$ and $L$ positive as well, conditions satisfied by hypothesis. This concludes the proof.

$\square$

Furthermore, let us note that both step size candidates, that are $\{\frac{2\mu + 2\sqrt{\mu L}}{L - \mu}, \frac{\mu + \gamma + \sqrt{(\mu - \gamma)^2 + 4\gamma L}}{L - \mu}\}$ respectively for the cases $\gamma = \mu$ and $\gamma > \mu$ show that NAG-GS method converges in the case $L \to \mu$ with a step size that tends to $\infty$, this behavior cannot be anticipated by the upper-bound given by (Luo & Chen, 2021, Theorem 1). Some simple numerical experiments are performed in Section 1.1.5 to support this theoretical result.

Finally, based on previous discussions, let us remark that for $\alpha \in [\frac{\mu + \gamma + \sqrt{(\mu - \gamma)^2 + 4\gamma L}}{L - \mu}, \infty]$ when $\gamma \neq \mu$ or $\alpha \in [\frac{2\mu + 2\sqrt{\mu L}}{L - \mu}, \infty]$ when $\gamma = \mu$, we have $\rho(E(\alpha)) = |\lambda_4(\alpha)|$ and one can show that $\rho(E)$ is strictly monotonically increasing function of $\alpha$ for all $L > \mu > 0$ and $\gamma > 0$, see Lemma 2 for the formal proof.

**Lemma 2.** *Given $\gamma > 0$, and assuming $0 < \mu < L$, then for $\gamma = \mu$ and $\gamma > \mu$, the spectral radius $\rho(E(\alpha))$ is a strict monotonic increasing function of $\alpha$ for $\alpha \in [\alpha_c, \infty]$ with $\alpha_c = \frac{2\mu + 2\sqrt{\mu L}}{L - \mu}$ or $\alpha_c = \frac{\mu + \gamma + \sqrt{(\mu - \gamma)^2 + 4\gamma L}}{L - \mu}$.*

*Proof.* Let us first recall that on $[\alpha_c, \infty]$, the spectral radius $\rho(E(\alpha))$ is equal to $|\lambda_4|$, the expression of $\lambda_4$ as a function of parameters of interests for the convergence analysis of NAG-GS method was given in (5) and recalled here-under for convenience:

$$
\lambda_4 = \frac{2\gamma + \alpha\gamma + \alpha\mu - L\alpha^2 + \alpha^2\mu}{2(\gamma + \alpha\gamma + \alpha\mu + \alpha^2\mu)} -
$$
$$
\frac{\alpha\sqrt{L^2\alpha^2 - 2L\alpha^2\mu - 2L\alpha\mu - 2\gamma L\alpha - 4\gamma L + \alpha^2\mu^2 + 2\alpha\mu^2 + 2\gamma\alpha\mu + \mu^2 + 2\gamma\mu + \gamma^2}}{2(\gamma + \alpha\gamma + \alpha\mu + \alpha^2\mu)}
\tag{7}
$$

Let start by showing that $\lambda_4$ is negative on $[\alpha_c, \infty]$. Firstly, one can easily observe that the denominator of $\lambda_4$ is positive, secondly let us compute the values for $\alpha$ such that:

$$
2\gamma + \alpha\gamma + \alpha\mu - L\alpha^2 + \alpha^2\mu -
$$
$$
\alpha\sqrt{L^2\alpha^2 - 2L\alpha^2\mu - 2L\alpha\mu - 2\gamma L\alpha - 4\gamma L + \alpha^2\mu^2 + 2\alpha\mu^2 + 2\gamma\alpha\mu + \mu^2 + 2\gamma\mu + \gamma^2} = 0
$$
$$
\equiv -4\gamma^2 - 4\alpha\gamma(\mu + \gamma) + \alpha^2(\gamma^2 - 4\gamma L + 2\gamma\mu + \mu^2) - \alpha^2(\gamma^2 - 4\gamma L + 6\gamma\mu + \mu^2) = 0
$$
$$
\equiv (-4\gamma\mu)\alpha^2 - 4\gamma(\mu + \gamma)\alpha - 4\gamma^2 = 0
\tag{8}
$$

The expression above is negative as soon as $\alpha < -1$ or $\alpha > \frac{-\gamma}{\mu} < 0$ since $\gamma, \mu > 0$ by hypothesis. The latter is always satisfied since $\alpha \geq \alpha_c > 0$ by hypothesis. Therefore $\rho(E(\alpha)) = -\lambda_4$ for $\alpha \in [\alpha_c, \infty]$.

To show the monotonic increasing behavior of $\rho(E(\alpha))$ w.r.t. $\alpha \in [\alpha_c, \infty]$, it remains to show that:

$$
\frac{d(\rho(E(\alpha))}{d\alpha} = \frac{d(-\lambda_4)}{d\alpha} > 0.
\tag{9}
$$

To ease the analysis, let us decompose $-\lambda_4(\alpha) = t_1(\alpha) + t_2(\alpha)$ such that:

$$
t_1(\alpha) = -\frac{2\gamma + \alpha\gamma + \alpha\mu - L\alpha^2 + \alpha^2\mu}{2(\gamma + \alpha\gamma + \alpha\mu + \alpha^2\mu)}
$$
$$
t_2(\alpha) = \frac{\alpha\sqrt{L^2\alpha^2 - 2L\alpha^2\mu - 2L\alpha\mu - 2\gamma L\alpha - 4\gamma L + \alpha^2\mu^2 + 2\alpha\mu^2 + 2\gamma\alpha\mu + \mu^2 + 2\gamma\mu + \gamma^2}}{2(\gamma + \alpha\gamma + \alpha\mu + \alpha^2\mu)}
\tag{10}
$$

Let us now show that $\frac{dt_1(\alpha)}{d\alpha} > 0$ and $\frac{dt_2(\alpha)}{d\alpha} > 0$ for any $L > \mu > 0$. We first obtain:

$$
\frac{dt_1(\alpha)}{d\alpha} = \frac{(2\gamma + 2\mu + 4\alpha\mu)(2\gamma + \alpha\gamma + \alpha\mu - L\alpha^2 + \alpha^2\mu)}{(2\gamma + 2\alpha\gamma + 2\alpha\mu + 2\alpha^2\mu)^2} -
$$
$$
\frac{\gamma + \mu - 2L\alpha + 2\alpha\mu}{2\gamma + 2\alpha\gamma + 2\alpha\mu + 2\alpha^2\mu}
$$
$$
= \frac{(L\alpha^2 + \gamma)(\gamma + \mu) + 2\alpha\gamma(L + \mu)}{2(\alpha + 1)^2(\gamma + \alpha\mu)^2}
\tag{11}
$$

which is strictly positive since $L > \mu > 0$ and $\gamma > 0$ by hypothesis. Furthermore:

$$
\frac{dt_2(\alpha)}{d\alpha} =
$$
$$
\frac{(\gamma + \mu)(L - \mu)(\alpha^3 L - 3\alpha\gamma) + \alpha^2(L(-\gamma^2 - \mu^2) + 2\gamma(L^2 - L\mu + \mu^2)) + \gamma(\gamma^2 - 2\gamma(2L - \mu) + \mu^2)}{2(\alpha + 1)^2(\alpha\mu + \gamma)^2\sqrt{\alpha^2(L^2 - 2L\mu + \mu^2) - 2\alpha(\gamma + \mu)(L - \mu) + \gamma^2 - 2\gamma(2L - \mu) + \mu^2}}
\tag{12}
$$

The remaining demonstration is significantly long and technically heavy in the case $\gamma > \mu$. Then we limit the last part of the demonstration for the case $\mu = \gamma$ for which we have shown previously

than $\alpha_c = \frac{\mu + \gamma + 2\sqrt{\gamma L}}{L - \mu} = \frac{2\mu + 2\sqrt{\mu L}}{L - \mu}$. In practice, with respect to the NAG-GS method summarized by Algorithm 1, $\gamma$ quickly decreases to $\mu$ and equality $\mu = \gamma$ holds for the most part of the iterations of the Algorithm, hence this case is more important to detail here. However, the reasoning explained herein ultimately leads to identical final conclusions when considering the case where $\gamma$ is greater than $\mu$.

The first term of the numerator of Equation 12 is positive as soon as $\alpha \geq \sqrt{\frac{3\gamma}{L}}$. In the case $\mu = \gamma$, we determine the conditions under which the second term of the numerator of Equation 12 is positive, that is:

$$
\begin{aligned}
&\alpha^2(L(-2\mu^2) + 2\mu(L^2 - L\mu + \mu^2)) + \mu(2\mu^2 - 2\mu(2L - \mu)) > 0 \\
\equiv\ &\alpha^2(L(-2\mu^2) + 2\mu(L^2 - L\mu + \mu^2)) > \mu(-2\mu^2 + 2\mu(2L - \mu))
\end{aligned}
\tag{13}
$$

First one can see that:

$$
\begin{aligned}
(L(-2\mu^2) + 2\mu(L^2 - L\mu + \mu^2)) &> 0, \\
\mu(-2\mu^2 + 2\mu(2L - \mu)) &> 0
\end{aligned}
\tag{14}
$$

hold as soon as $L > \mu > 0$ which is satisfied by hypothesis. Therefore, the second term of the numerator of Equation 12 is positive as soon as

$$
\alpha > \sqrt{\frac{\mu(-2\mu^2 + 2\mu(2L - \mu))}{(L(-2\mu^2) + 2\mu(L^2 - L\mu + \mu^2))}} = \sqrt{\frac{2\mu}{L - \mu}}
\tag{15}
$$

which exists since $L > \mu > 0$ by hypothesis (the second root of (14) being negative). Finally, since $\alpha \in [\alpha_c, \infty]$ by hypothesis, $\frac{dt_2(\alpha)}{d\alpha}$ is positive as soon as:

$$
\begin{aligned}
\alpha_c &> \sqrt{\frac{3\mu}{L}} \\
\alpha_c &> \sqrt{\frac{2\mu}{L - \mu}}
\end{aligned}
\tag{16}
$$

hold with $\alpha_c = \frac{2\mu + 2\sqrt{\mu L}}{L - \mu}$. One can easily show that both inequalities hold as soon as $L > \mu > 0$ which is satisfied by the hypothesis. This concludes the proof of the strict increasing monotonicity of $\rho(E(\alpha))$ w.r.t. $\alpha$ for $\alpha \in [\alpha_c, \infty]$ assuming $L > \mu > 0$ and $\gamma = \mu$.

$\square$

### 1.1.2 COVARIANCE ANALYSIS

In this section, we study the contribution to the computation of maximum step size for the NAG-GS method through the analysis of the covariance matrix at stationarity. Let us start by computing the covariance matrix $C$ obtained at iteration $k + 1$ from Algorithm 1:

$$
C_{k+1} = \mathbb{E}(y_{k+1} y_{k+1}^T)
\tag{17}
$$

By denoting $\xi_k = \begin{bmatrix} 0 \\ \sigma \frac{\sqrt{\alpha}}{1+\tau} \eta_k \end{bmatrix}$, let us replace $y_{k+1}$ by its expression given in (3), (17) writes:

$$
\begin{aligned}
C_{k+1} &= \mathbb{E}(y_{k+1} y_{k+1}^T) \\
&= \mathbb{E}\left((Ey_k + \xi_k)(Ey_k + \xi_k)^T\right) \\
&= \mathbb{E}\left(Ey_k y_k^T E^T\right) + \mathbb{E}\left(\xi_k \xi_k^T\right)
\end{aligned}
\tag{18}
$$

which holds since expectation operator $\mathbb{E}(.)$ is a linear operator and by assuming statistical independence between $\xi_k$ and $Ey_k$. On the one hand, by using again the properties of linearity of $\mathbb{E}$ and since $E$ is seen as a constant by $\mathbb{E}(.)$, one can show that $\mathbb{E}\left(Ey_k y_k^T E^T\right) = EC_k E^T$. On the other hand, since $\eta_k \sim \mathcal{N}(0, 1)$, then Equation (18) becomes:

$$
C_{k+1} = EC_k E^T + Q
\tag{19}
$$

where $Q = \begin{bmatrix} 0_{2\times2} & 0_{2\times2} \\ 0_{2\times2} & \frac{\alpha_k \sigma^2}{(1+\tau_k)^2} I_{2\times2} \end{bmatrix}$. Let us now look at the limiting behavior of Equation (19), that is $\lim_{k\to\infty} C_k$. Let be $C = \lim_{k\to\infty} C_k$ the covariance matrix reached in the asymptotic regime, also referred to as stationary regime. Applying the limit on both sides of Equation (19), $C$ then satisfies

$$C = ECE^T + Q \tag{20}$$

Hence (20) is a particular case of discrete Lyapunov equation. For solving such equation, the vectorization operator denoted $\vec{\cdot}$ is applied on both sides on (20), this amounts to solve the following linear system:

$$(I_{4^2\times4^2} - E \otimes E)\vec{C} = \vec{Q} \tag{21}$$

where $A \otimes B = \begin{bmatrix} a_{11}B & \cdots & a_{1n}B \\ \vdots & \ddots & \vdots \\ a_{m1}B & \cdots & a_{mn}B \end{bmatrix}$ stands for the Kronecker product. The solution is given by:

$$C = \overleftarrow{(I_{4^2\times4^2} - E \otimes E)^{-1}\vec{Q}} \tag{22}$$

where $\overleftarrow{a}$ stands for the un-vectorized operator.

Let us note that, even for the 2-dimensional case considered in this section, the dimension of matrix $C$ rapidly growth and cannot be written in plain within this paper. For the following, we will keep its symbolic expression. The stationary matrix $C$ quantifies the spreading of the limit of the sequence $\{y_k\}$, as a direct consequence of the Brownian motion effect. Now we look at the directions that maximize the scattering of the points, in other words, we are looking for the eigenvectors and the associated eigenvalues of $C$. Actually, the required information for the analysis of the step size is contained within the expression of the eigenvalues $\lambda_i(C)$. The obtained eigenvalues are rationale functions w.r.t. the parameters of the schemes, while their numerator brings less interest for us (supported further), we will focus on their denominator. We obtained the following expressions:

$$\lambda_1(C) = \frac{N_1(\alpha,\mu,L,\gamma,\sigma)}{D_1(\alpha,\mu,L,\gamma,\sigma)},$$
$$\text{s.t. } D_1(\alpha,\mu,L,\gamma,\sigma) = -L^2\alpha^3\mu - L^2\alpha^2\mu - \gamma L^2\alpha^2 + 2L\alpha^3\mu^2 + 4L\alpha^2\mu^2 +$$
$$4\gamma L\alpha^2\mu + 2L\alpha\mu^2 + 8\gamma L\alpha\mu + 2\gamma^2 L\alpha + 4\gamma L\mu + 4\gamma^2 L \tag{23}$$

$$\lambda_2(C) = \frac{N_2(\alpha,\mu,L,\gamma,\sigma)}{D_2(\alpha,\mu,L,\gamma,\sigma)},$$
$$\text{s.t. } D_2(\alpha,\mu,L,\gamma,\sigma) = \alpha^3\mu^3 + 3\alpha^2\mu^3 + 3\gamma\alpha^2\mu^2 + 2\alpha\mu^3 +$$
$$8\gamma\alpha\mu^2 + 2\gamma^2\alpha\mu + 4\gamma\mu^2 + 4\gamma^2\mu \tag{24}$$

$$\lambda_3(C) = \frac{N_3(\alpha,\mu,L,\gamma,\sigma)}{D_3(\alpha,\mu,L,\gamma,\sigma)},$$
$$\text{s.t. } D_3(\alpha,\mu,L,\gamma,\sigma) = \alpha^3\mu^3 + 3\alpha^2\mu^3 + 3\gamma\alpha^2\mu^2 + 2\alpha\mu^3 +$$
$$8\gamma\alpha\mu^2 + 2\gamma^2\alpha\mu + 4\gamma\mu^2 + 4\gamma^2\mu \tag{25}$$

$$\lambda_4(C) = \frac{N_4(\alpha,\mu,L,\gamma,\sigma)}{D_4(\alpha,\mu,L,\gamma,\sigma)},$$
$$\text{s.t. } D_4(\alpha,\mu,L,\gamma,\sigma) = -L^2\alpha^3\mu - L^2\alpha^2\mu - \gamma L^2\alpha^2 + 2L\alpha^3\mu^2 + 4L\alpha^2\mu^2 +$$
$$4\gamma L\alpha^2\mu + 2L\alpha\mu^2 + 8\gamma L\alpha\mu + 2\gamma^2 L\alpha + 4\gamma L\mu + 4\gamma^2 L \tag{26}$$

One can observe that:

1. Given $\alpha, L, \mu, \gamma$ positive, the denominators of eigenvalues $\lambda_2$ and $\lambda_3$ are positive as well, unlike eigenvalues $\lambda_1$ and $\lambda_4$ for which some vertical asymptotes may appear. The latter will be studied in more detail further. Note that, even if some eigenvalues share the same denominator, it is not the case for the numerator. This will be illustrated later in Figures 5 and 6 to ease the analysis.

2. Interestingly, the volatility of the noise defined by the parameter $\sigma$ does not appear within the expressions of the denominators. It gives us a hint that these vertical asymptotes are due to the fact that spectral radius is getting close to 1 (discussed further in Section 1.1.3). Moreover, the parameter $\sigma$ appears only within the numerators and based on intensive numerical tests, this parameter has a pure scaling effect onto the eigenvalues $\lambda_i(C)$ when studied w.r.t. $\alpha$ without modifying the trends of the curves.

Let us now study in more details the denominator of $\lambda_1$ and $\lambda_4$ and seek for critical step size as a function of $\gamma, \mu$ and $L$ at which a vertical asymptote may appear by solving:

$$
\begin{aligned}
&- L^2\alpha^3\mu - L^2\alpha^2\mu - \gamma L^2\alpha^2 + 2L\alpha^3\mu^2 + 4L\alpha^2\mu^2 + \\
&4\gamma L\alpha^2\mu + 2L\alpha\mu^2 + 8\gamma L\alpha\mu + 2\gamma^2 L\alpha + 4\gamma L\mu + 4\gamma^2 L = 0 \\
&\equiv \mu(2\mu - L)\alpha^3 + (\mu + \gamma)(4\mu - L)\alpha^2 + (2\mu^2 + 8\gamma\mu + 2\gamma^2)\alpha + 4\gamma(\mu + \gamma) = 0
\end{aligned}
\tag{27}
$$

This polynomial equation in $\alpha$ has three roots:

$$
\begin{aligned}
\alpha_1 &= \frac{-\gamma - \mu}{\mu}, \\
\alpha_2 &= \frac{\mu + \gamma - \sqrt{\gamma^2 - 6\gamma\mu + \mu^2 + 4\gamma L}}{L - 2\mu}, \\
\alpha_3 &= \frac{\mu + \gamma + \sqrt{\gamma^2 - 6\gamma\mu + \mu^2 + 4\gamma L}}{L - 2\mu}.
\end{aligned}
\tag{28}
$$

First, it is obvious that the first root $\alpha_1$ is negative given $\gamma, \mu$ assumed nonnegative and therefore can be disregarded. Concerning $\alpha_2$ and $\alpha_3$, those are real roots as soon as:

$$
\begin{aligned}
&\gamma^2 - 6\gamma\mu + \mu^2 + 4\gamma L \geq 0 \\
&\equiv (\gamma - \mu)^2 - 4\gamma\mu + 4\gamma L \geq 0 \\
&\equiv (\gamma - \mu)^2 \geq 4\gamma(\mu - L)
\end{aligned}
\tag{29}
$$

which is always satisfied since $\gamma > 0$ and $0 < \mu < L$ by hypothesis.

Further, it is obvious that the study must include three scenarios:

1. Scenario 1: $L - 2\mu < 0$, or equivalently $\mu > L/2$. Given $\mu$ and $\gamma$ positive by hypothesis, it implies that $\alpha_3$ is negative and hence can be disregarded. It remains to check if $\alpha_2$ can be positive, it amounts to verifying if

$$
\begin{aligned}
&\mu + \gamma - \sqrt{\gamma^2 - 6\gamma\mu + \mu^2 + 4\gamma L} < 0 \\
&\equiv (\mu + \gamma)^2 < \gamma^2 - 6\gamma\mu + \mu^2 + 4\gamma L \\
&\equiv \mu < \frac{L}{2}
\end{aligned}
$$

which never holds by hypothesis. Therefore, for the first scenario, there is no positive critical step size at which a vertical asymptote for the eigenvalues may appear.

2. Scenario 2: $L - 2\mu > 0$, or equivalently $\mu < L/2$. Obviously, $\alpha_3$ is positive and hence shall be considered for the analysis of maximum step size for our NAG-GS method. It remains to check if $\alpha_2$ is positive, that is to verify if the numerator can be negative. We have seen in the first scenario that $\alpha_2$ is negative as soon as $\mu < \frac{L}{2}$ which is verified by hypothesis. Therefore, only $\alpha_3$ is positive.

3. Scenario 3: $L - 2\mu = 0$. For such a situation, the critical step size is located at $\infty$ and can be disregarded as a potential limitation in our study.

In summary, a potentially critical and limiting step size only exists in the case $\mu < L/2$, or equivalently if $\kappa(A) > 2$. In this setting, the critical step size is positive and is equal to $\alpha_{\text{crit}} = \frac{\mu + \gamma + \sqrt{\gamma^2 - 6\gamma\mu + \mu^2 + 4\gamma L}}{L - 2\mu}$. Figures 3 to 4 display the evolution of the eigenvalues $\lambda_i(C)$ for $1 \leq i \leq 4$ w.r.t. to $\alpha$ for the two first scenarios, that are for $\mu > L/2$ and $\mu < L/2$. For the first scenario, the parameters $\sigma, \gamma, \mu$ and $L$ have been respectively set to $\{1, 3/2, 1, 3/2\}$. For the

second scenario, $\sigma$, $\gamma$, $\mu$ and $L$ have been respectively set to $\{1, 3/2, 1, 3\}$. As expected, one can observe in Figure 3 that no vertical asymptote is present. Furthermore, one can observe $\lambda_i(C)$ seem to converge to some limit point when $\alpha \to \infty$, numerically we report that this limit point is zero, for all the values of $\gamma$ and $\sigma$ considered.

Finally, again as expected by the results presented in this section, Figure 4 shows the presence of two vertical asymptotes for the eigenvalues $\lambda_1$ and $\lambda_4$, and none for $\lambda_2$ and $\lambda_3$. Moreover, the critical step size is approximately located at $\alpha = 6$ which aligns with analytical expression $\alpha_{\text{crit}} = \frac{\mu + \gamma + \sqrt{\gamma^2 - 6\gamma\mu + \mu^2 + 4\gamma L}}{L - 2\mu}$. Finally, one can observe that, after the vertical asymptotes, all the eigenvalues converge to some limit points, again numerically we report that this limit point is zero, for all the values of $\gamma$ and $\sigma$ considered.

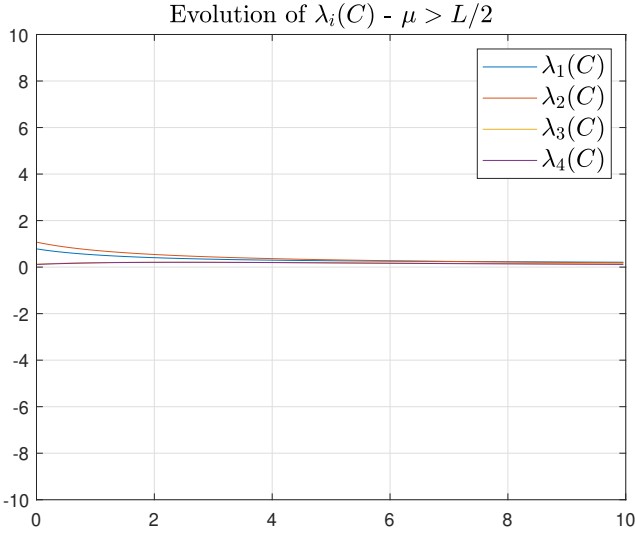

Figure 3: Evolution of $\lambda_i(C)$ w.r.t $\alpha$ for scenario $\mu > L/2$; $\sigma = 1$, $\gamma = 3/2$, $\mu = 1$, $L = 3/2$.

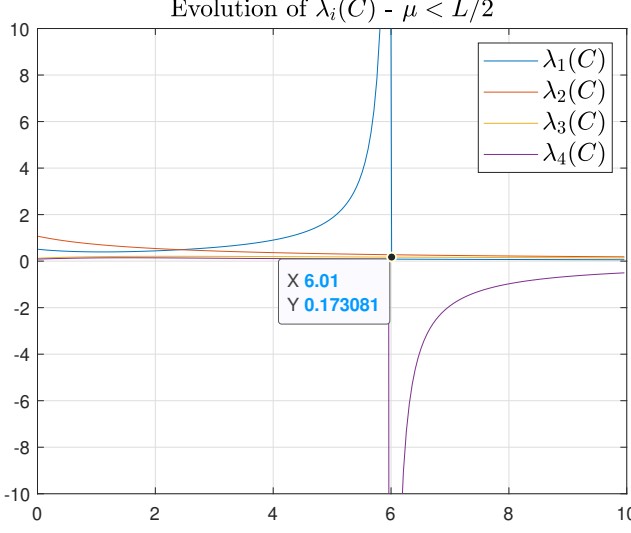

Figure 4: Evolution of $\lambda_i(C)$ w.r.t $\alpha$ for scenario $\mu < L/2$; $\sigma = 1$, $\gamma = 3/2$, $\mu = 1$, $L = 3$.

### 1.1.3 A CONCLUSION FOR THE 2-DIMENSIONAL CASE

In Section 1.1.1 and Section 1.1.2, several theoretical results have been derived for coming up with appropriate choices of constant step size for Algorithm 1. Key insights and interesting values for the step size have been discussed from the study of the spectral radius of iteration matrix $E$ and through the analysis of the covariance matrix in the asymptotic regime. Let us summarize the theoretical results obtained:

- from the spectral radius analysis of iteration matrix $E$; two scenarios have been highlighted, that are:

  1. case $\gamma = \mu$: the step size $\alpha$ that minimizes the spectral radius of matrix $E$ is $\alpha = \frac{2\mu + 2\sqrt{\mu L}}{L - \mu}$,

  2. case $\gamma > \mu$: the step size $\alpha$ that minimizes the spectral radius of matrix $E$ is $\alpha = \frac{\mu + \gamma + \sqrt{(\mu - \gamma)^2 + 4\gamma L}}{L - \mu}$.

- from the analysis of covariance matrix $C$ at stationarity: in the case $L - 2\mu > 0$, or equivalently $\mu < L/2$, we have seen that there is a vertical asymptote for two eigenvalues of $C$ at $\alpha_{\text{crit}} = \frac{\mu + \gamma + \sqrt{\gamma^2 - 6\gamma\mu + \mu^2 + 4\gamma L}}{L - 2\mu}$, leading to an intractable scattering of the limit points $\{y_k\}_{k\to\infty}$ generated by Algorithm 1. In the case $\mu > L/2$, there is no positive critical step size at which a vertical asymptote for the eigenvalues may appear.

Therefore, for quadratic functions such that $\mu > L/2$, we can safely choose either $\alpha = \frac{2\mu + 2\sqrt{\mu L}}{L - \mu}$ when $\gamma = \mu$ either $\alpha = \frac{\mu + \gamma + \sqrt{(\mu - \gamma)^2 + 4\gamma L}}{L - \mu}$ when $\gamma > \mu$ to get the minimal spectral radius for iteration matrix $E$ and hence the highest contraction rate for the NAG-GS method.

For quadratic functions such that $\mu < L/2$, we must show that the NAG-GS method is stable for both step sizes. Let us denote by $\alpha_c = \{\frac{2\mu + 2\sqrt{\mu L}}{L - \mu}, \frac{\mu + \gamma + \sqrt{(\mu - \gamma)^2 + 4\gamma L}}{L - \mu}\}$, two values of step size for the two scenarios $\gamma = \mu$ and $\gamma > \mu$. In Lemma 3, we show that NAG-GS is asymptotically convergent, or stable, for the 2-dimensional case under mild assumptions in the case $\mu < L/2$.

**Lemma 3.** *Given $\gamma > 0$, and assuming $0 < \mu < L/2$, then for $\gamma = \mu$ and $\gamma > \mu$ the following inequalities respectively hold:*

$$\frac{\mu + \gamma + \sqrt{\gamma^2 - 6\gamma\mu + \mu^2 + 4\gamma L}}{L - 2\mu} > \frac{2\mu + 2\sqrt{\mu L}}{L - \mu}$$
$$\frac{\mu + \gamma + \sqrt{\gamma^2 - 6\gamma\mu + \mu^2 + 4\gamma L}}{L - 2\mu} > \frac{\mu + \gamma + \sqrt{(\mu - \gamma)^2 + 4\gamma L}}{L - \mu} \tag{30}$$

*Thus, in the 2-dimensional case, NAG-GS is asymptotically convergent (or stable) when choosing $\alpha_c = \frac{\mu + \gamma + \sqrt{(\mu - \gamma)^2 + 4\gamma L}}{L - \mu}$ or $\alpha_c = \frac{2\mu + 2\sqrt{\mu L}}{L - \mu}$ respectively for the cases $\gamma > \mu$ and $\gamma = \mu$.*

*Proof.* In order to prove the asymptotic stability or convergence of NAG-GS for the 2-dimensional case within the set of assumptions detailed above, one must show that $\rho(E(\alpha_c)) < 1$ for the two choices of $\alpha_c$.

Let us start by computing $\alpha$ such that $\rho(E(\alpha)) = 1$. As proved in Lemma 2, for $\alpha \in [\alpha_c, \infty]$, $\rho(E(\alpha)) = -\lambda_4$ with $\lambda_4$ given in (5), we then have to compute $\alpha$ such that:

$$-\lambda_4 = -\frac{2\gamma + \alpha\gamma + \alpha\mu - L\alpha^2 + \alpha^2\mu}{2(\gamma + \alpha\gamma + \alpha\mu + \alpha^2\mu)} +$$
$$\frac{\alpha(L^2\alpha^2 - 2L\alpha^2\mu - 2L\alpha\mu - 2\gamma L\alpha - 4\gamma L + \alpha^2\mu^2 + 2\alpha\mu^2 + 2\gamma\alpha\mu + \mu^2 + 2\gamma\mu + \gamma^2)^{1/2}}{2(\gamma + \alpha\gamma + \alpha\mu + \alpha^2\mu)} = 1.$$

This leads to computing the roots of a quadratic polynomial equation in $\alpha$, the positive root is:

$$\alpha = \frac{\gamma + \mu + \sqrt{4L\gamma + \gamma^2 - 6\gamma\mu + \mu^2}}{L - 2\mu} \tag{31}$$

which not surprisingly identifies to $\alpha_{\text{crit}}$ from the covariance matrix analysis [1].

Furthermore, as per Lemma 2, $\rho(E(\alpha))$ is strictly monotonically increasing function over the interval $[\alpha_c, \infty]$. Therefore, showing that $\rho(E(\alpha_c)) < 1$ is equivalent to show that $\alpha_c$ is strictly lower than $\alpha_{\text{crit}} := \frac{\gamma + \mu + \sqrt{4L\gamma + \gamma^2 - 6\gamma\mu + \mu^2}}{L - 2\mu}$.

Let us focus on the case $\gamma > \mu$; since $0 < \mu < L/2$ by hypothesis, the second inequality from (30) can be written as:

$$(L - \mu)(\gamma + \mu + \sqrt{(\gamma - \mu)^2 + 4\gamma(L - \mu)}) - (L - 2\mu)(\gamma + \mu + \sqrt{(\gamma - \mu)^2 + 4\gamma L}) > 0$$
$$\equiv \gamma\mu + \mu^2 + (L - \mu)\sqrt{\gamma^2 + \mu^2 + \gamma(4L - 6\mu)} + (2\mu - L)\sqrt{(\gamma - \mu)^2 + 4\gamma L} > 0$$

Given $\gamma, \mu > 0$, it remains to show that:

$$(L - \mu)\sqrt{\gamma^2 + \mu^2 + \gamma(4L - 6\mu)} + (2\mu - L)\sqrt{(\gamma - \mu)^2 + 4\gamma L} > 0 \tag{32}$$

In order to show this, we study the conditions for $\gamma$ such that the left-hand side of (32) is positive. With simple manipulations, one can show that canceling the left-hand side of (32) boils down to canceling the following quadratic polynomial:

$$(L - \mu)\sqrt{\gamma^2 + \mu^2 + \gamma(4L - 6\mu)} + (2\mu - L)\sqrt{(\gamma - \mu)^2 + 4\gamma L} = 0$$
$$\equiv (-3\mu + 2L)\gamma^2 + (2\mu^2 - 8L\mu + 4L^2)\gamma + 2L\mu^2 - 3\mu^3 = 0$$

The two roots are:

$$\gamma_1 = \frac{-\mu^2 - 2L^2 - 2\sqrt{-2\mu^4 + L^4 - 4\mu L^3 + 4\mu^2 L^2 + \mu^3 L} + 4\mu L}{2L - 3\mu}$$
$$\gamma_2 = \frac{-\mu^2 - 2L^2 + 2\sqrt{-2\mu^4 + L^4 - 4\mu L^3 + 4\mu^2 L^2 + \mu^3 L} + 4\mu L}{2L - 3\mu},$$

which are real and distinct as soon as:

$$-2\mu^4 + L^4 - 4\mu L^3 + 4\mu^2 L^2 + \mu^3 L > 0$$
$$\equiv (L - 2\mu)(L - \mu)(-\mu^2 + L^2 - \mu L) > 0,$$

which holds since $0 < \mu < L/2$ by hypothesis (one can easily show that $-\mu^2 + L^2 - \mu L$ is positive in such setting). Moreover, the denominator $2L - 3\mu$ is strictly positive since $0 < \mu < L/2$. One can check that $\gamma_1$ is negative for all $\gamma, L > 0$ and $0 < \mu < L/2$ (simply show that $-\mu^2 - 2L^2 + 4\mu L$ is negative) and can be disregarded since $\gamma$ is positive by hypothesis. Therefore, proving that (32) holds is equivalent to show that:

$$\gamma > \frac{-\mu^2 - 2L^2 + 2\sqrt{(L - 2\mu)(L - \mu)(-\mu^2 + L^2 - \mu L)} + 4\mu L}{2L - 3\mu} \tag{33}$$

To achieve this, let us first show that

$$\mu > \frac{-\mu^2 - 2L^2 + 2\sqrt{(L - 2\mu)(L - \mu)(-\mu^2 + L^2 - \mu L)} + 4\mu L}{2L - 3\mu}$$
$$\equiv 0 > \mu^2 + \sqrt{(L - 2\mu)(L - \mu)(-\mu^2 + L^2 - \mu L)} - L^2 + \mu L$$
$$\equiv -\mu^2 + L^2 - \mu L > (L - 2\mu)(L - \mu)$$
$$\equiv \mu < \frac{2}{3}L,$$

which holds by hypothesis. Since $\gamma > \mu$ by hypothesis, inequality (33) holds for any $\mu$ and $L$ positive as well, conditions satisfied by hypothesis.

Finally, since $\frac{\mu + \gamma + \sqrt{(\mu - \gamma)^2 + 4\gamma L}}{L - \mu} > \frac{\mu + \gamma + 2\sqrt{\gamma L}}{L - \mu}$ for any $\gamma, \mu, L > 0$, then first inequality in (30) holds as well. This concludes the proof. $\square$

---

[1]It explains why the critical $\alpha$ does not include $\sigma$, this singularity is due to the spectral radius reaching the value 1.

We conclude this section by discussing several important insights:

- Except for $\alpha_{\text{crit}}$, we do not report significant information coming from the analysis of $\lambda_i(C)$ for the computation of the step size and the validity of the candidates for $\alpha$ that are from $\left\{\frac{2\mu+2\sqrt{\mu L}}{L-\mu}, \frac{\mu+\gamma+\sqrt{(\mu-\gamma)^2+4\gamma L}}{L-\mu}\right\}$ respectively for the cases $\gamma = \mu$ and $\gamma > \mu$.

- Concerning the effect of the volatility $\sigma$ of the noise, we have mentioned earlier that the parameter $\sigma$ appears only within the numerators $\lambda_i(C)$ and based on intensive numerical tests, this parameter has a pure scaling effect onto the eigenvalues $\lambda_i(C)$ when studied w.r.t. $\alpha$ without modifying the trends of the curves. For compliance purpose, Figures 5 and 6 respectively show the evolution of the numerators $N_i(\alpha, \mu, L, \gamma, \sigma)$ of eigenvalues expressions of $C$ given in Equations (23) to (26) w.r.t. $\sigma$, for both scenarios $\mu < L/2$ and $\mu > L/2$. One can observe monotonic polynomial increasing behavior of $N_i(\alpha, \mu, L, \gamma, \sigma)$ w.r.t $\sigma$ for all $1 \leq i \leq 4$.

- The theoretical analysis summarized in this section is valid for the 2-dimensional case, we show in Section 1.1.4 how to generalize our results for the $n$-dimensional case. This has no impact on our results.

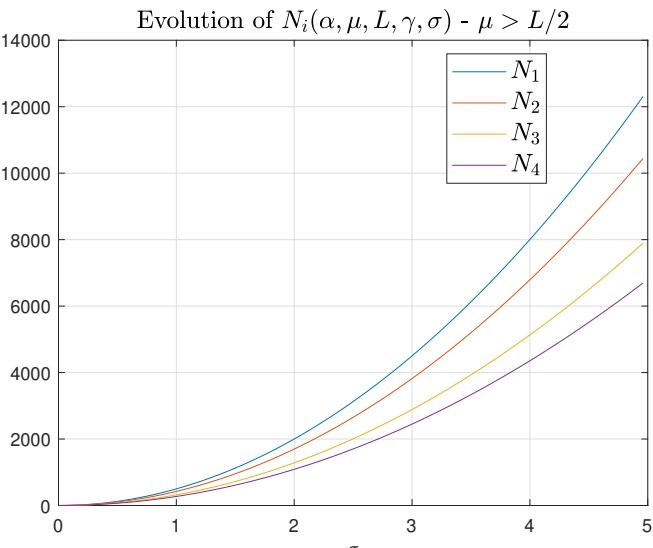

Figure 5: Evolution of $N_i(\alpha, \mu, L, \gamma, \sigma)$ w.r.t $\sigma$ for scenario $\mu > L/2$; $\gamma = 3/2$, $\mu = 1$, $L = 3/2$, $\alpha = \frac{\mu+\gamma+\sqrt{(\mu-\gamma)^2+4\gamma L}}{L-\mu}$.

### 1.1.4 EXTENSION TO $n$-DIMENSIONAL CASE

In this section, we show that we can easily extend the results obtained for the 2-dimensional case in Section 1.1.1, Section 1.1.2 and Section 1.1.3 to the $n$-dimensional case with $n > 2$. Let us start by recalling that for NAG transformation (7), the general SDE's system to solve for the quadratic case is:

$$\dot{y}(t) = \begin{bmatrix} -I_{n\times n} & I_{n\times n} \\ 1/\gamma(\mu I_{n\times n} - A) & -\mu/\gamma I_{n\times n} \end{bmatrix} y(t) + \begin{bmatrix} 0_{n\times 1} \\ \frac{dZ}{dt} \end{bmatrix}, \quad t > 0. \tag{34}$$

Let recall that $y = (x, v)$ with $x, v \in \mathbb{R}^n$, let $n$ be even and let consider the permutation matrix $P$ associated to permutation indicator $\pi$ given here-under in two-line form:

$$\pi = \begin{bmatrix} (1 & 2) & (3 & 4) & \cdots & (n-1 & n) & (n+1 & n+2) & \cdots & (2n-1 & 2n) \\ (2*1-1 & 2*1) & (2*3-1 & 2*3) & \cdots & (2n-3 & 2n-2) & (3 & 4) & \cdots & (2n-1 & 2n) \end{bmatrix}$$

where the bottom second-half part of $\pi$ corresponds to the complementary of the bottom first half w.r.t. to the set $\{1, 2, ..., 2n\}$ in the increasing order. For avoiding ambiguities, the ones element of $P$ are at indices $(\pi(1, j), \pi(2, j))$ for $1 \leq j \leq 2n$. For such convention and since permutation matrix

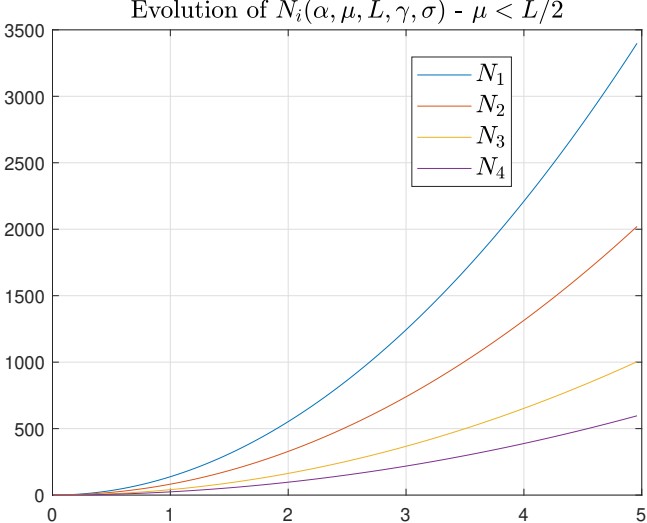

Figure 6: Evolution of $N_i(\alpha, \mu, L, \gamma, \sigma)$ w.r.t $\sigma$ for scenario $\mu < L/2$; $\gamma = 3/2$, $\mu = 1$, $L = 3$, $\alpha = \frac{\mu + \gamma + \sqrt{(\mu - \gamma)^2 + 4\gamma L}}{L - \mu}$.

$P$ associated to indicator $\pi$ is orthogonal matrix, (34) can be equivalently written as follows:

$$
\begin{aligned}
\dot{y}(t) &= PP^T \begin{bmatrix} -I_{n \times n} & I_{n \times n} \\ 1/\gamma(\mu I_{n \times n} - A) & -\mu/\gamma I_{n \times n} \end{bmatrix} PP^T y(t) + \begin{bmatrix} 0_{n \times 1} \\ \dot{Z} \end{bmatrix}, \\
&\equiv P^T \dot{y}(t) = P^T \begin{bmatrix} -I_{n \times n} & I_{n \times n} \\ 1/\gamma(\mu I_{n \times n} - A) & -\mu/\gamma I_{n \times n} \end{bmatrix} PP^T y(t) + P^T \begin{bmatrix} 0_{n \times 1} \\ \dot{Z} \end{bmatrix},
\end{aligned}
\tag{35}
$$

Since we assumed w.l.o.g. $A = \text{diag}(\lambda_1, \ldots, \lambda_n)$ with $\mu = \lambda_1 \leq \ldots \leq \lambda_n = L$, one can easily see that Equation (35) has the structure:

$$
\begin{bmatrix} \dot{x}_1 \\ \dot{x}_2 \\ \dot{v}_1 \\ \dot{v}_2 \\ \vdots \\ \dot{x}_{2i-1} \\ \dot{x}_{2i} \\ \dot{v}_{2i-1} \\ \dot{v}_{2i} \\ \vdots \\ \dot{x}_{n-1} \\ \dot{x}_n \\ \dot{v}_{n-1} \\ \dot{v}_n \end{bmatrix} = \begin{bmatrix} \begin{matrix} I_2 & -I_2 \\ 1/\gamma(\mu I_2 - A_1) & -\mu/\gamma I_2 \end{matrix} & 0 & 0 & 0 & 0 \\ 0 & \ddots & 0 & 0 & 0 \\ 0 & 0 & \begin{matrix} I_2 & -I_2 \\ 1/\gamma(\mu I_2 - A_i) & -\mu/\gamma I_2 \end{matrix} & 0 & 0 \\ 0 & 0 & 0 & \ddots & 0 \\ 0 & 0 & 0 & 0 & \begin{matrix} I_2 & -I_2 \\ 1/\gamma(\mu I_2 - A_m) & -\mu/\gamma I_2 \end{matrix} \end{bmatrix} \cdot \begin{bmatrix} x_1 \\ x_2 \\ v_1 \\ v_2 \\ \vdots \\ x_{2i-1} \\ x_{2i} \\ v_{2i-1} \\ v_{2i} \\ \vdots \\ x_{n-1} \\ x_n \\ v_{n-1} \\ v_n \end{bmatrix} + \begin{bmatrix} 0 \\ 0 \\ \dot{Z}_1 \\ \dot{Z}_2 \\ \vdots \\ 0 \\ 0 \\ \dot{Z}_{2i-1} \\ \dot{Z}_{2i} \\ \vdots \\ 0 \\ 0 \\ \dot{Z}_{n-1} \\ \dot{Z}_n \end{bmatrix}
\tag{36}
$$

which boils down to $m = \frac{n}{2}$ independent 2-dimensional SDE's systems where $A_i = \text{diag}(\lambda_{2i-1}, \lambda_{2i})$ with $1 \leq i \leq m$ such that $\lambda_1 = \mu$ and $\lambda_n = L$.

Therefore, the $m$ SDE's systems can be studied and theoretically solved independently with the schemes and the associated step sizes presented in previous sections. However, in practice, we will use a unique and general step size $\alpha$ to tackle the full SDE's system 34.

Let now use the "decoupled" structure given in (36) to come up with a general step size that will ensure the convergence of each system and hence the convergence of the full original system given in (34). Let us denote by $\alpha_i$ the step size for the $i$-th SDE's system with $1 \leq i \leq m = n/2$ minimizing the spectral radius of the system at hand. For convenience, let us consider the case $\gamma > \mu$, we apply the same method as detailed in Section 1.1.1 and Section 1.1.2 to compute the expression of $\alpha_i$ that minimizes $\rho(E_i(\alpha))$, we obtain:

$$
\alpha_i = \frac{\mu + \gamma + \sqrt{(\mu - \gamma)^2 + 4\gamma\lambda_{2i}}}{\lambda_{2i} - \mu}
\tag{37}
$$

Finally, in Theorem 1, we show that choosing $\alpha_c := \alpha = \frac{\mu+\gamma+\sqrt{(\mu-\gamma)^2+4\gamma L}}{L-\mu}$ ensures the convergence of NAG-GS method used to solve the SDE's system 34 in the $n$-dimensional case for $n > 2$. Theorem 1 is enunciated in Section 2.3 in the main text and the proof is given here-under.

*Proof.* First, we recall that Lemma 3 in Section 1.1.3 provides the proof for the asymptotic convergence of NAG-GS method for $n = 2$ when choosing $\alpha := \alpha_c = \frac{\mu+\gamma+\sqrt{(\mu-\gamma)^2+4\gamma L}}{L-\mu}$ for the case $\gamma > \mu$. In particular, it is shown that the spectral radius of the iteration matrix $\rho(E(\alpha_c))$ is strictly lower than 1 under consistent assumptions with the ones of Theorem 1 (see Lemma 3 for more details). The following steps of the proof show that choosing $\alpha_c$ also leads to the asymptotic convergence of NAG-GS method for $n > 2$.

To do so, let us start by considering, w.l.o.g., the SDE's system in the form given by (36) and let $\alpha_i = \frac{\mu+\gamma+\sqrt{(\mu-\gamma)^2+4\gamma\lambda_{2i}}}{\lambda_{2i}-\mu}$ be the step size (given in Equation (37)) selected for solving the $i$-th SDE's system with $1 \leq i \leq m = n/2$, minimizing $\rho(E_i(\alpha))$, that is the spectral radius of the associated iteration matrix $E_i$. The result of Lemma 3 can be directly extended for each independent 2-dimensional SDE's system, in particular showing that $\rho(E_i(\alpha_i)) < 1$ for $1 \leq i \leq m = n/2$.

Therefore, to prove the convergence of the NAG-GS method by choosing a single step size $\alpha$ such that $0 < \alpha \leq \frac{\mu+\gamma+\sqrt{(\mu-\gamma)^2+4\gamma L}}{L-\mu}$, it suffices to show that:

$$\alpha = \frac{\mu + \gamma + \sqrt{(\mu - \gamma)^2 + 4\gamma L}}{L - \mu} \leq \min_{1 \leq i \leq m = n/2} \alpha_i \tag{38}$$

For proving that (38) holds, it sufficient to show that for any $\lambda$ such that $0 < \mu \leq \lambda \leq L < \infty$ we have:

$$\frac{\mu + \gamma + \sqrt{(\mu - \gamma)^2 + 4\gamma L}}{L - \mu} \leq \frac{\mu + \gamma + \sqrt{(\mu - \gamma)^2 + 4\gamma\lambda}}{\lambda - \mu}. \tag{39}$$

which is equivalent to showing:

$$\frac{\mu + \gamma + \sqrt{(\mu - \gamma)^2 + 4\gamma L}}{L - \mu} - \frac{\mu + \gamma + \sqrt{(\mu - \gamma)^2 + 4\gamma\lambda}}{\lambda - \mu} \leq 0$$

$$\equiv \gamma(\frac{1}{L-\mu} - \frac{1}{\lambda-\mu}) + \mu(\frac{1}{L-\mu} - \frac{1}{\lambda-\mu}) + \frac{\sqrt{(\mu-\gamma)^2+4\gamma L}}{L-\mu} - \frac{\sqrt{(\mu-\gamma)^2+4\gamma\lambda}}{\lambda-\mu} \leq 0 \tag{40}$$

Since $0 < \mu \leq \lambda \leq L < \infty$ by hypothesis, one can easily show that first two terms of the last inequality are negative. It remains to show that:

$$\frac{\sqrt{(\mu-\gamma)^2+4\gamma L}}{L-\mu} - \frac{\sqrt{(\mu-\gamma)^2+4\gamma\lambda}}{\lambda-\mu} \leq 0$$

$$\equiv (-\gamma^2 - 4\gamma\lambda + 2\gamma\mu - \mu^2)L^2 + (4\gamma\lambda^2 + 2\gamma^2\mu + 2\mu^3)L + \tag{41}$$

$$\gamma^2\lambda^2 - 2\gamma^2\lambda\mu - 2\gamma\lambda^2\mu + \lambda^2\mu^2 - 2\lambda\mu^3 \leq 0$$

Note that we can easily show that the coefficient of $L^2$ is negative, hence last inequality is satisfied as soon as $L \leq \frac{-\gamma^2\lambda+2\gamma^2\mu+2\gamma\lambda\mu-\lambda\mu^2+2\mu^3}{\gamma^2+4\gamma\lambda-2\gamma\mu+\mu^2}$ or $L \geq \lambda$. The latter condition is satisfied by hypothesis, this concludes the proof.

Note that one can check that $\frac{-\gamma^2\lambda+2\gamma^2\mu+2\gamma\lambda\mu-\lambda\mu^2+2\mu^3}{\gamma^2+4\gamma\lambda-2\gamma\mu+\mu^2} \leq \lambda$. $\qquad\square$

The theoretical results derived in these sections along with the key insights are validated in Section 1.1.5 through numerical experiments conducted for the NAG-GS method in the quadratic case.

### 1.1.5  NUMERICAL TESTS FOR QUADRATIC CASE

In this section, we report some simple numerical tests for the NAG-GS method (Algorithm 1) used to tackle the accelerated SDE's system given in (11) where:

- the objective function is $f(x) = (x - ce)^T A(x - ce)$ with $A \in \mathbb{S}_+^3$, $e$ a all-ones vector of dimension 3 and $c$ a positive scalar. For such a strongly convex setting, since the feasible set is $V = \mathbb{R}^3$, the minimizer $\arg\min f$ uniquely exists and is simply equal to $ce$; it will be denoted further by $x^\star$. The matrix $A$ is generated as follows: $A = QAQ^{-1}$ where matrix $D$ is a diagonal matrix of size 3 and $Q$ is a random orthogonal matrix. This test procedure allows us to specify the minimum and maximum eigenvalues of $A$ that are respectively $\mu$ and $L$ and hence it allows us to consider the two scenarios discussed in Section 1.1.1, that are $\mu > L/2$ and $\mu < L/2$.
- The noise volatility $\sigma$ is set to 1, we report that this corresponds to a significant level of noise.
- Initial parameter $\gamma_0$ is set to $\mu$.
- Different values for the step size $\alpha$ will be considered in order to empirically demonstrate the optimal choice $\alpha_c$ in terms of contraction rate, but also validate the critical values for step size in the case $\mu < L/2$ and, finally, highlight the effect of the step size in terms of scattering of the final iterates generated by NAG-GS around the minimizer of $f$.

From a practical point of view, we consider $m = 200000$ points. For each of them, the NAG-GS method is run for a maximum number of iterations to reach the stationarity, and the initial state $x_0$ is generated using normal Gaussian distribution. Since $f(x)$ is a quadratic function, it is expected that the points will converge to some Gaussian distribution around the minimizer $x^\star = ce$. Furthermore, since the initial distribution is also Gaussian, then it is expected that the intermediate distributions (at each iteration of the NAG-GS method) are Gaussian as well. Therefore, in order to quantify the rate of convergence of the NAG-GS method for different values of step size, we will monitor $\|\bar{x}^k - x^\star\|$, that is the distance between the empirical mean of the distribution at iteration $k$ and the minimizer $x^\star$ of $f$.

Figures 7 and 8 respectively show the evolution of $\|\bar{x}^k - x^\star\|$ along iteration and the final distribution of points obtained by NAG-GS at stationarity for the scenario $\mu > L/2$, for the latter the points are projected onto the three planes to have a full visualization. As expected by the theory presented in Section 1.1.3, there is no critical $\alpha$, hence one may choose arbitrary large values for step size while the NAG-GS method still converges. Moreover, the choice of $\alpha = \alpha_c$ gives the highest rate of convergence. Finally, one can observe that the distribution of limit points tightens more and more around the minimizer $x^\star$ of $f$ as the chosen step increases, as expected by the analysis of Figure 3. Hence, one may choose a very large step size $\alpha$ so that the limit points converge to $x^\star$ almost surely but at a cost of a (much) slower convergence rate. Here comes the tradeoff between the convergence rate and the limit points scattering.

Finally, Figures 12 and 10 provide similar results for the scenario $\mu < L/2$. The theory outlined in Section 1.1.3 and Section 1.1.4 predicts a critical value of $\alpha$ that indicates when the convergence of NAG-GS is destroyed in such a scenario. In order to illustrate this gradually, different values of $\alpha$ have been chosen within the set $\{\alpha_c, \alpha_c/2, (\alpha_c + \alpha_{\text{crit}})/2, 0.98\alpha_{\text{crit}}\}$. First, one can observe that the choice of $\alpha = \alpha_c$ gives again the highest rate of convergence, see Figure 12. Moreover, one can clearly see that for $\alpha \to \alpha_{\text{crit}}$, the convergence starts to fail and the spreading of the limit points tends to infinity. We report that for $\alpha = \alpha_{\text{crit}}$, NAG-GS method diverges. Again, these numerical results are fully predicted by the theory derived in previous sections.

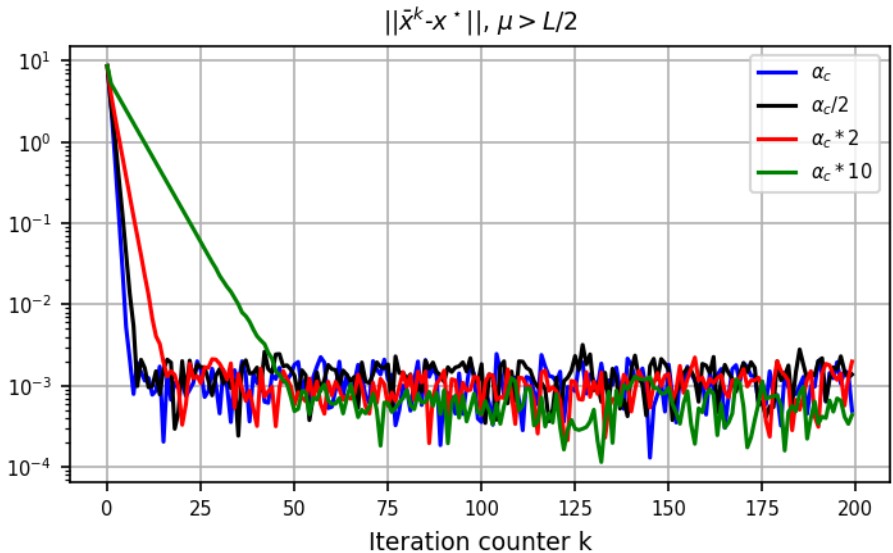

Figure 7: Evolution of $\|\bar{x}^k - x^\star\|$ along iteration for the scenario $\mu > L/2$; $c = 5$, $\gamma = \mu = 1$, $L = 1.9$ and $\sigma = 1$ for $\alpha \in \{\alpha_c, \alpha_c/2, 2\alpha_c, 10\alpha_c\}$ with $\alpha_c = \frac{2\mu + 2\sqrt{\mu L}}{L - \mu} = 5.29$.

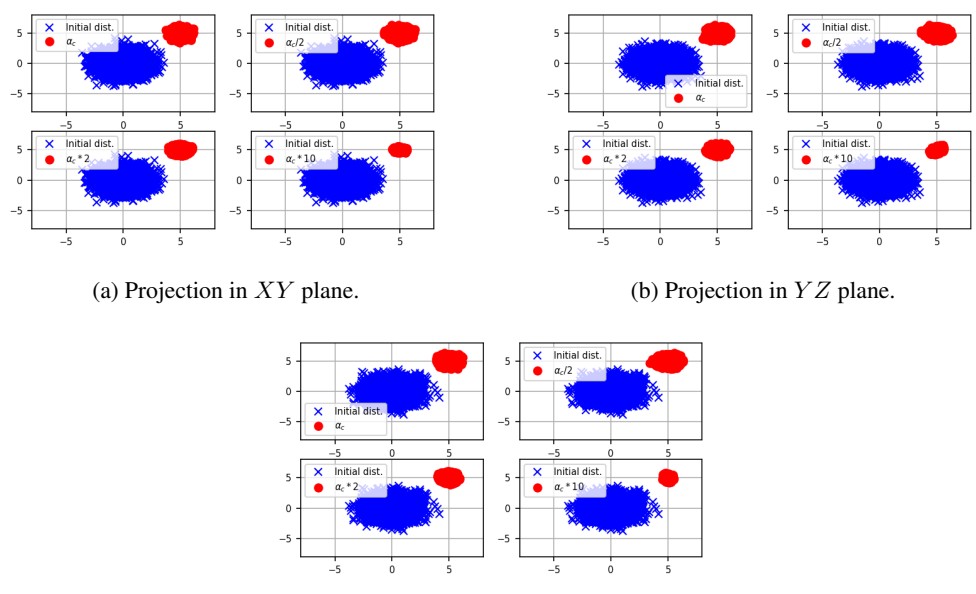

(a) Projection in $XY$ plane.

(b) Projection in $YZ$ plane.

(c) Projection in $XZ$ plane.

Figure 8: Initial (blue crosses) and final (red circles) distributions of points generated by the NAG-GS method for the scenario $\mu > L/2$; $c = 5$, $\gamma = \mu = 1$, $L = 1.9$ and $\sigma = 1$ for $\alpha \in \{\alpha_c, \alpha_c/2, 2\alpha_c, 10\alpha_c\}$ with $\alpha_c = \frac{2\mu + 2\sqrt{\mu L}}{L - \mu} = 5.29$.

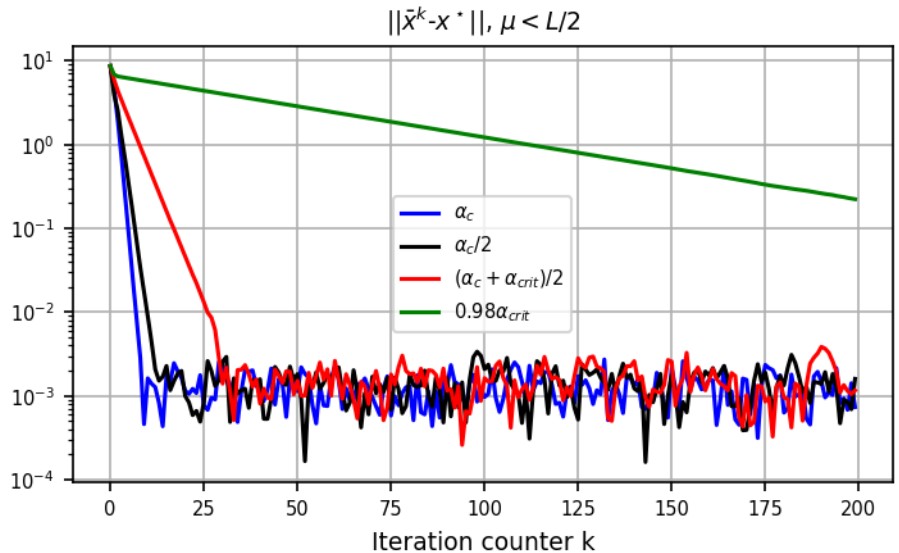

Figure 9: Evolution of $\|\bar{x}^k - x^\star\|$ along iteration for the scenario $\mu < L/2$; $c = 5$, $\gamma = \mu = 1$, $L = 3$ and $\sigma = 1$ for $\alpha \in \{\alpha_c, \alpha_c/2, (\alpha_c + \alpha_{\text{crit}})/2, 0.98\alpha_{\text{crit}}\}$ with $\alpha_c = \frac{2\mu + 2\sqrt{\mu L}}{L - \mu} = 2.73$ and $\alpha_{\text{crit}} = \frac{\mu + \gamma + \sqrt{\gamma^2 - 6\gamma\mu + \mu^2 + 4\gamma L}}{L - 2\mu} = 4.83$.

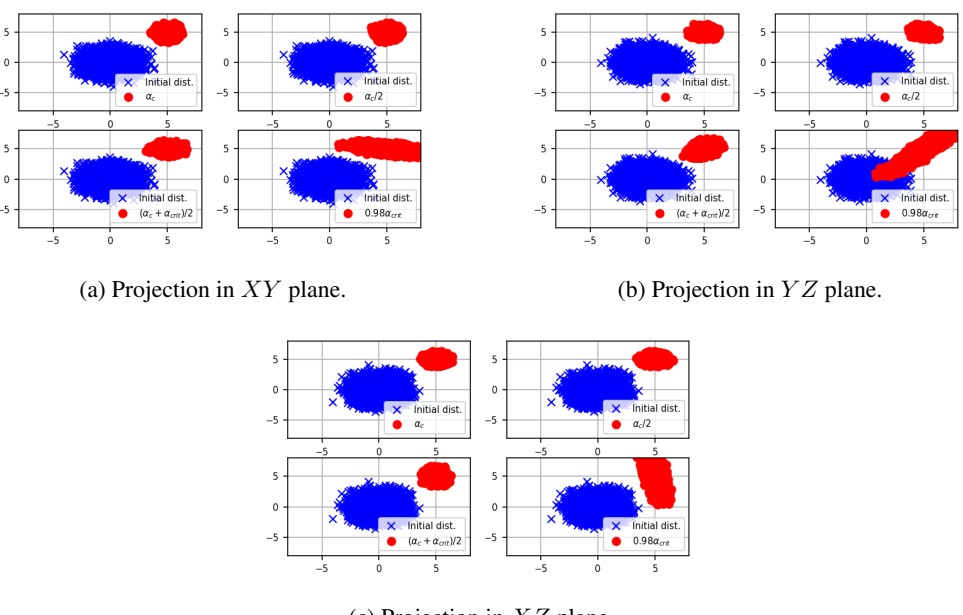

Figure 10: Initial (blue crosses) and final (red circles) distributions of points generated by the NAG-GS method for scenario $\mu < L/2$; $c = 5$, $\gamma = \mu = 1$, $L = 3$ and $\sigma = 1$ for $\alpha \in \{\alpha_c, \alpha_c/2, (\alpha_c + \alpha_{\text{crit}})/2, 0.98\alpha_{\text{crit}}\}$ with $\alpha_c = \frac{2\mu + 2\sqrt{\mu L}}{L - \mu} = 2.73$ and $\alpha_{\text{crit}} = \frac{\mu + \gamma + \sqrt{\gamma^2 - 6\gamma\mu + \mu^2 + 4\gamma L}}{L - 2\mu} = 4.83$.

## 1.2 FULLY-IMPLICIT SCHEME

In this section, we present an iterative method based on the NAG transformation $G_{NAG}$ (7) along with a fully implicit discretization to tackle (4) in the stochastic setting, the resulting method shall be

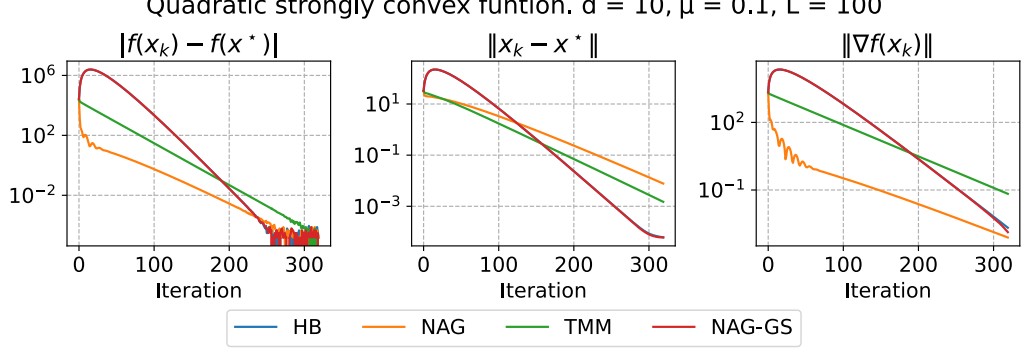

Figure 11: Comparison of different accelerated first order methods for strongly convex quadratics objective with dimension $d = 10$, strong convexity constant $\mu = 1$ and smoothness constant $L = 20$. **HB** - Heavy Ball Polyak (1964), **NAG** - Nesterov Accelerated Gradient Nesterov (1983), **TMM** - Triple Momentum Method Van Scoy et al. (2017), **NAG-GS** - Nesterov Accelerated Gradient with Gauss-Seidel Splitting. All methods were set with optimal hyperparameters.

Figure 12: Comparison of different accelerated first order methods for strongly convex quadratics objective with dimension $d = 10$, strong convexity constant $\mu = 0.1$ and smoothness constant $L = 100$. **HB** - Heavy Ball Polyak (1964), **NAG** - Nesterov Accelerated Gradient Nesterov (1983), **TMM** - Triple Momentum Method Van Scoy et al. (2017), **NAG-GS** - Nesterov Accelerated Gradient with Gauss-Seidel Splitting. All methods were set with optimal hyperparameters.

referred to as "NAG-FI" method. We propose the following discretization for (6) perturbated with noise; given step size $\alpha_k > 0$:

$$
\begin{aligned}
\frac{x_{k+1} - x_k}{\alpha_k} &= v_{k+1} - x_{k+1}, \\
\frac{v_{k+1} - v_k}{\alpha_k} &= \frac{\mu}{\gamma_k}(x_{k+1} - v_{k+1}) - \frac{1}{\gamma_k}Ax_{k+1} + \sigma\frac{W_{k+1} - W_k}{\alpha_k}.
\end{aligned}
\tag{42}
$$

As done for the NAG-GS method, from a practical point of view, we will use $W_{k+1} - W_k = \Delta W_k = \sqrt{\alpha_k}\eta_k$ where $\eta_k \sim \mathcal{N}(0, 1)$, by the properties of the Brownian motion.

In the quadratic case, that is $f(x) = \frac{1}{2}x^\top Ax$, solving (42) is equivalent to solve:

$$
\begin{bmatrix} x_k \\ v_k + \sigma\sqrt{\alpha_k}\eta_k \end{bmatrix} = \begin{bmatrix} (1 + \alpha_k)I & -\alpha_k I \\ \frac{\alpha_k}{\gamma_k}(A - \mu I) & (1 + \frac{\alpha_k\mu}{\gamma_k})I \end{bmatrix} \begin{bmatrix} x_{k+1} \\ v_{k+1} \end{bmatrix}
\tag{43}
$$

where $\eta_k \sim \mathcal{N}(0, 1)$. Furthermore, ODE (8) from the main text is again discretized implicitly:

$$
\frac{\gamma_{k+1} - \gamma_k}{\alpha_k} = \mu - \gamma_{k+1}, \quad \gamma_0 > 0.
\tag{44}
$$

As done for NAG-GS method, heuristically, for general $f \in \mathcal{S}_{L,\mu}^{1,1}$ with $\mu \geq 0$, we just replace $Ax_{k+1}$ in (42) with $\nabla f(x_{k+1})$ and obtain the following NAG-FI scheme:

$$
\begin{aligned}
\frac{x_{k+1} - x_k}{\alpha_k} &= v_{k+1} - x_{k+1}, \\
\frac{v_{k+1} - v_k}{\alpha_k} &= \frac{\mu}{\gamma_k}(x_{k+1} - v_{k+1}) - \frac{1}{\gamma_k}\nabla f(x_{k+1}) + \sigma \frac{W_{k+1} - W_k}{\alpha_k}.
\end{aligned}
\tag{45}
$$

From the first equation, we get $v_{k+1} = \frac{x_{k+1} - x_k}{\alpha_k} + x_{k+1}$ that we substitute within the second equation, we obtain:

$$
x_{k+1} = \frac{v_k + \tau_k x_k - \frac{\alpha_k}{\gamma_k}\nabla f(x_{k+1}) + \sigma\sqrt{\alpha_k}\eta_k}{1 + \tau_k}
\tag{46}
$$

with $\tau_k = 1/\alpha_k + \mu/\gamma_k$.

Computing $x_{k+1}$ is equivalent to computing a fixed point of the operator given by the right-hand side of (46). Hence, it is also equivalent to finding the root of the function:

$$
g(u) = u - \left( \frac{v_k + \tau_k x_k - \frac{\alpha_k}{\gamma_k}\nabla f(u) + \sigma\sqrt{\alpha_k}\eta_k}{1 + \tau_k} \right)
\tag{47}
$$

with $g : \mathbb{R}^n \to \mathbb{R}^n$. In order to compute the root of this function, we consider a classical Newton-Raphson procedure detailed in Algorithm 2. In Algorithm 2, $J_g(.)$ denotes the Jacobian operator of

---

**Algorithm 2** Newton-Raphson method

---

**Input:** Choose the point $u_0 \in \mathbb{R}^n$, some $\alpha_k, \gamma_k, \tau_k > 0$.
    **for** $i = 0, 1, \dots$ **do**
        Compute $J_g(u_i) = I_n + \frac{\alpha_k}{\gamma_k(1+\tau_k)}\nabla^2 f(u_i)$
        Compute $g(u_i)$ using (47)
        Set $u_{i+1} = u_i - [J_g(u_i)]^{-1}g(u_i)$
    **end for**

---

function $g$ (47) w.r.t. $u$, $I_n$ denotes the identity matrix of size $n$ and $\nabla^2 f$ denotes the Hessian matrix of objective function $f$. Please note that the iterative method outlined in Algorithm 2 exhibits a connection to the family of second-order methods called the Levenberg-Marquardt algorithm Levenberg (1944); Marquardt (1963) applied to the unconstrained minimization problem $\min_{x \in \mathbb{R}^n} f(x)$ for a twice-differentiable function $f$. Finally, Algorithm 3 summarizes the NAG-FI method.

---

**Algorithm 3** NAG-FI Method

---

**Input:** Choose the point $x_0 \in \mathbb{R}^n$, set $v_0 = x_0$, some $\sigma \geq 0, \mu \geq 0, \gamma_0 > 0$.
    **for** $k = 0, 1, \dots$ **do**
        Sample $\eta_k \sim \mathcal{N}(0, 1)$
        Choose $\alpha_k > 0$
        Set $\gamma_{k+1} := \frac{\gamma_k + \alpha_k \mu}{1 + \alpha_k}$
        Set $\tau_{k+1} = 1/\alpha_k + \mu/\gamma_{k+1}$
        Compute the root $u$ of (47) by using Algorithm 2
        Set $x_{k+1} = u$
    **end for**

---

By following a similar stability analysis as the one performed for NAG-GS, one can show that this method is unconditionally A-stable as expected by the theory of implicit schemes. In particular, one can show that eigenvalues of the iterations matrix are positive decreasing functions w.r.t. step size $\alpha$, allowing then the choice of any positive value for $\alpha$. Similarly, one can show that the eigenvalues of the covariance matrix at stationarity associated with the NAG-FI method are decreasing functions w.r.t. $\alpha$ that tend to 0 as soon as $\alpha \to \infty$. It implies that Algorithm 3 is theoretically able to generate iterates that converge to $\arg \min f$ almost surely, even in the stochastic setting with the potentially quadratic rate of converge. This theoretical result is quickly highlighted in Figure 13 that shows the

final distribution of points generated by NAG-FI once used in test setup detailed in Section 1.1.5, in the most interesting and critical scenario $\mu < L/2$. As expected, $\alpha$ can be chosen as large as desired, we choose here $\alpha = 1000\alpha_c$. Moreover, for increasing $\alpha$, the final distributions of points are more and more concentrated around $x^*$.

Therefore, the NAG-FI method constitutes a good basis for deriving efficient second-order methods for tackling stochastic optimization problems, which is hard to find in the current SOTA. Indeed, second-order methods and more generally some variants of preconditioned gradient methods have recently been proposed and used in the deep learning community for the training of NN for instance. However, it appears that there is limited empirical success for such methods when used for training NN when compared to well-tuned Stochastic Gradient Descent schemes, see for instance Botev et al. (2017); Zeiler (2012). To the best of our knowledge, no theoretical explanations have been brought to formally support these empirical observations. This will be part of our future research directions.

Besides these nice preliminary theoretical results and numerical observations for small dimension problems, there is a limitation of the NAG-FI method that comes from the numerical feasibility for computing the root of the non-linear function (47) that can be very challenging in practice. We will try to address this issue in future works.

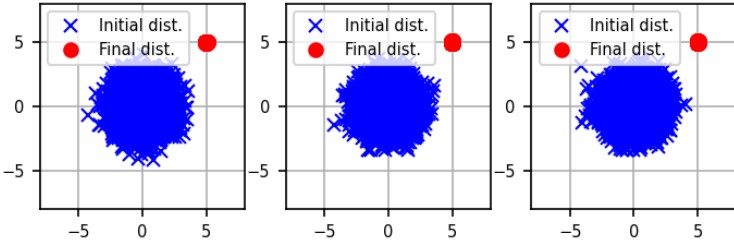

Figure 13: Projection onto $XY$, $YZ$, and $XZ$ planes (from left to right) of initial (blue crosses) and final (red circles) distributions of points generated by NAG-FI method - scenario $\mu < L/2$; $c = 5$, $\gamma = \mu = 1$, $L = 3$ and $\sigma = 1$ for $\alpha = 1000\alpha_c$ with $\alpha_c = \frac{2\mu + 2\sqrt{\mu L}}{L - \mu} = 2.73$.

## 1.3  ADDITIONAL INSIGHTS ABOUT THE NOTION A-STABILITY

In this section we recall the concept of A-stability of ODE solvers, which is the classical notion of "negative real part" by Dahlquistit Dahlquist (1963). First we note that the discussion about A-stability of solver for general ODE in the form $\dot{x}(t) = f(t, x(t))$ with $x(0) = x_0$, $\Re(\lambda) < 0 \forall \lambda \in \sigma(J_f)$ can be long and tedious, hence we consider a simple linear ODE of the form

$$\dot{x}(t) = Gx(t), \quad x(0) = x_0 \quad \text{with} \quad \Re(\lambda) < 0 \ \forall \lambda \in \sigma(G). \tag{48}$$

A one-step method for solving ODE (48) with step size $\alpha > 0$ can be written as $x_{k+1} = E(G, \alpha)x_k$. The numerical scheme is called absolute stable or A-stable if $\rho(E(G, \alpha)) < 1$ (from which the asymptotic convergence $x_k \to 0$ follows). If $\rho(E(G, \alpha)) < 1$ holds for all $\alpha > 0$, then the scheme is called unconditionally A-stable, and if $\rho(E(G, \alpha)) < 1$ holds for some $\alpha \in I$, where $I$ denotes an interval of the positive half line, then the scheme is conditionally A-stable. In the next subsection, we consider two popular schemes on the point of view of A-stability.

### 1.3.1  EXPLICIT AND IMPLICIT EULER SCHEMES

Here we review the stability of the explicit and implicit Euler schemes for solving Eq.(48). The analytical solution for Eq.(48) with a constant discretization step $\alpha$ generates the iterates:

$$x_{k+1} = x_k + \int_{t_k}^{t_{k+1}} Gx(s)ds, \quad k = 0, \dots, M - 1.$$

For $G = -A$, the explicit Euler method approximates the integral by the area of a rectangle with width $\alpha$ and height $-Ax_k$. This leads to the iterates $x_{k+1} = (I - \alpha A)x_k$ which corresponds to the GD scheme for minimizing $\Phi(x) = \frac{1}{2}x^T Ax$. The explicit Euler method is A-stable if the spectral radius of $I - \alpha A$ is strictly less than 1, i.e., if $\rho(I - \alpha A) = \max_{\lambda \in \sigma(I - \alpha A)} |\lambda| < 1$. We can easily show that $\rho(I - \alpha A) = \max(|1 - \alpha\mu|, |1 - \alpha L|)$, where $\mu$ and $L$ respectively denote the smallest and largest eigenvalue of $A$. Therefore, the explicit Euler method is A-stable if $0 < \alpha < 2/L$. Additionally, we can determine the optimal $\alpha$ that minimizes the spectral radius: $\min_{\alpha>0} \rho(I - \alpha A)$, which gives $\alpha^\star = 2/(\mu + L)$, resulting in $\rho(I - \alpha^\star A) = (Q_f - 1)/(Q_f + 1)$. Assuming $0 < \mu \le L < \infty$, we have $0 < \alpha^\star = 2/(\mu + L) < 2/L$. Hence, the explicit Euler method is A-stable, and the norm convergence with a linear rate follows as in Eq.(49).

$$\|x_k - x^\star\| \ \le \ \left(\frac{Q_f - 1}{Q_f + 1}\right)^k \|x_0 - x^\star\| \quad \text{and} \quad \Phi(x_k) - f^\star \le \frac{L}{2}\left(\frac{Q_f - 1}{Q_f + 1}\right)^{2k} \|x_0 - x^\star\|^2. \tag{49}$$

On the other hand, the implicit Euler scheme approximates the integral by the area of a rectangle with a height of $-Ax_{k+1}$, leading to the iterates $x_{k+1} = (I + \alpha A)^{-1}x_k$. The term $\rho(I + \alpha A)^{-1}$ can be expressed as $\max\left(|\frac{1}{1+\alpha\mu}|, |\frac{1}{1+\alpha L}|\right)$. This implies that the stability condition $\rho(I + \alpha A)^{-1} < 1$ holds true for all $\alpha > 0$, making the implicit Euler scheme unconditionally A-stable. Moreover, the implicit Euler method can achieve a faster convergence rate by time rescaling, as it is not limited by any constraints on the step size. This is equivalent to opting for a larger step size.

## 2 CONVERGENCE TO THE STATIONARY DISTRIBUTION

Another way to study the convergence of the proposed algorithms is to consider the Fokker-Planck equation for the density function $\rho(t, x)$. We will consider the simple case of the scalar SDE for the stochastic gradient flow (similarly as in (11)). Here $f : \mathbb{R} \to \mathbb{R}$:

$$dx = -\nabla f(x)dt + dZ = -\nabla f(x)dt + \sigma dW, \quad x(0) \sim \rho(0, x).$$

It is well known, that the density function for $x(t) \sim \rho(t, x)$ satisfies the corresponding Fokker-Planck equation:

$$\frac{\partial \rho(t, x)}{\partial t} = \nabla \left( \rho(t, x)\nabla f(x) \right) + \frac{\sigma^2}{2}\Delta \rho(t, x) \tag{50}$$

For the (50) one could write down the stationary (with $t \to \infty$) distribution

$$\rho^*(x) = \lim_{t \to \infty} \rho(t, x) = \frac{1}{Z} \exp\left( -\frac{2}{\sigma^2} f(x) \right), \quad Z = \int_{x \in V} \exp\left( -\frac{2}{\sigma^2} f(x) \right) dx. \tag{51}$$

It is useful to compare different optimization algorithms in terms of convergence in the probability space because it allows us to study the methods in the non-convex setting. We have to address two problems with this approach. Firstly, we need to specify some distance functional between current distribution $\rho_t = \rho(t, x)$ and stationary distribution $\rho^* = \rho^*(x)$. Secondly, we do not need to have access to the densities $\rho_t, \rho^*$ themselves.

For the first problem, we will consider the following distance functionals between probability distributions in the scalar case:

- **Kullback-Leibler divergence.** Several studies dedicated to convergence in probability space are available Arnold et al. (2001); Chewi et al. (2020); Lambert et al. (2022). We used the approach proposed in Pérez-Cruz (2008) to estimate KL divergence between continuous distributions based on their samples.

- **Wasserstein distance.** Wasserstein distance is relatively easy to compute for scalar densities. Also, it was shown, that the stochastic gradient process with a constant learning rate is exponentially ergodic in the Wasserstein sense Latz (2021).

- **Kolmogorov-Smirnov statistics.** We used the two-sample Kolmogorov-Smirnov test for goodness of fit.

To the best of our knowledge, the explicit formula for the stationary distribution of Fokker-Planck equations for the ASG SDE (11) remains unknown. That is why we have decided to get samples from the empirical stationary distributions using Euler-Maruyama integration Maruyama (1955) with a small enough step size of corresponding SDE with a bunch of different independent initializations.

We tested two functions, which are presented in Figure 14. We initially generated 100 points uniformly in the function domain. Then we independently solved the initial value problem (9) for each of them with Maruyama (1955). Results of the integration are presented in Figure 15. One can see, that in the relatively easy case (Figure 14a), NAG-GS converges faster, than gradient flow to its stationary distribution, see Figure 15a. At the same time, in the hard case (Figure 14b), NAG-GS is more robust to the large step size, see Figure 15b.

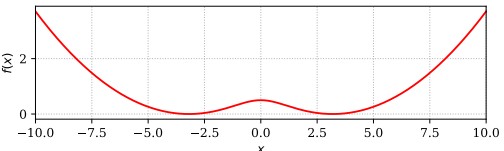

(a) Two pits function.
$f_1(x) = \frac{1}{50} \left( 2\log\left(\cosh(x)\right) - 5 \right)^2$

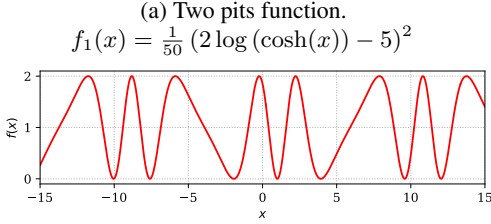

(b) Frequently modulated sin function.
$f_2(x) = \cos\left( 1.6x + \frac{5}{3}\sin(0.64x) - \pi \right)$

Figure 14: Non convex scalar functions to test

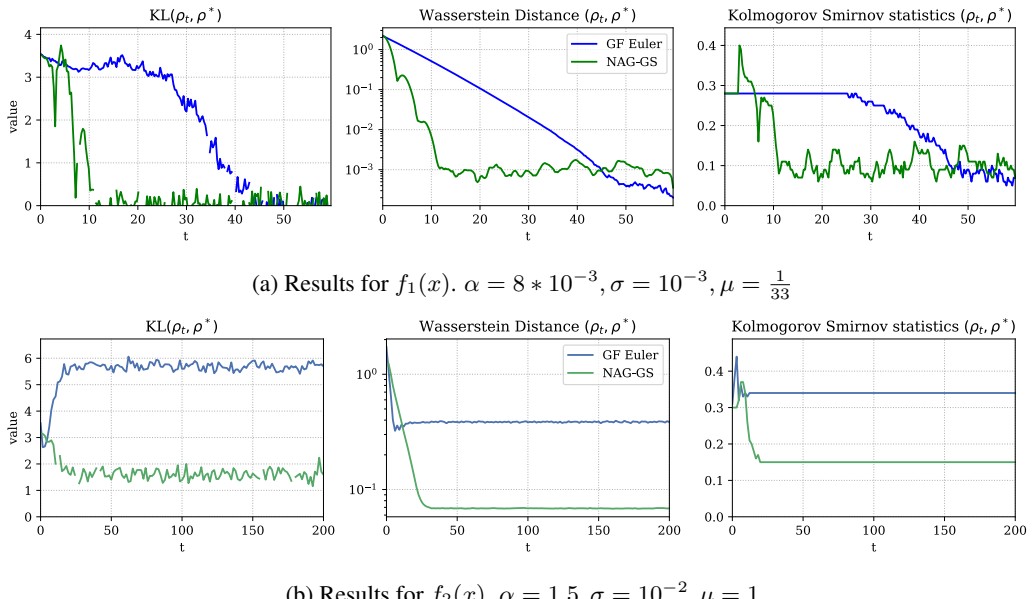

(a) Results for $f_1(x)$. $\alpha = 8 * 10^{-3}, \sigma = 10^{-3}, \mu = \frac{1}{33}$

(b) Results for $f_2(x)$. $\alpha = 1.5, \sigma = 10^{-2}, \mu = 1$

Figure 15: Convergence in probabilities of Euler integration of Gradient Flow (GF Euler) and NAG-GS for the non-convex scalar problems.

## 3  ADDITIONAL INSIGHTS

In this section, we provide additional experimental details. In particular, we discuss a little bit more our experimental setup and give some insights about NAG-GS as well.

Our computational resources are limited to a single Nvidia DGX-1 with 8 GPUs Nvidia V100. Almost all experiments were carried out on a single GPU. The only exception is for the training of ResNet50 on ImageNet which used all 8 GPUs.

### 3.1  PHASE DIAGRAMS

In Section 3.4 we mentioned that the lowest eigenvalues $\mu$ of approximated Hessian matrices evaluated during the training of the ResNet-20 model were negative. Furthermore, our theoretical analysis of NAG-GS in the convex case includes some conditions on the optimizer parameters $\alpha$, $\gamma$, and $\mu$. In particular, it is required that $\mu > 0$ and $\gamma \geq \mu$. In order to bring some insights about these remarks in the non-convex setting and inspired by Velikanov et al. (2022), we experimentally study the convergence regions of NAG-GS and sketch out the phase diagrams of convergence for different projection planes, see Figure 16.

We consider the same setup as in Section 3.4 in the main text, a paragraph about the ResNet-20 model, and use hyper-optimization library OPTUNA Akiba et al. (2019). Our preliminary experiments on RoBERTa show that $\alpha$ should be of magnitude $10^{-1}$. With the estimate of the Hessian spectrum of ResNet-20, we define the following search space

$$\alpha \sim \text{LogUniform}(10^{-2}, 10^2), \quad \gamma \sim \text{LogUniform}(10^{-2}, 10^2), \quad \mu \sim \text{Uniform}(-10, 100).$$

We sample a fixed number of triples and train the ResNet-20 model on CIFAR-10. The objective function is a top-1 classification error.

We report that there is a convergence almost everywhere within the projected search space onto $\alpha$-$\gamma$ plane (see Figure 16). The analysis of projections onto $\alpha$-$\mu$ and $\gamma$-$\mu$ planes brings different conclusions: there are regions of convergence for negative $\mu$ for some $\alpha < \alpha_{th}$ and $\gamma > \gamma_{th}$. Also, there is a subdomain of negative $\mu$ comparable to a domain of positive $\mu$ in the sense of the target metrics. Moreover, the majority of sampled points are located in the vicinity of the band $\lambda_{\min} < \mu < \lambda_{\max}$.

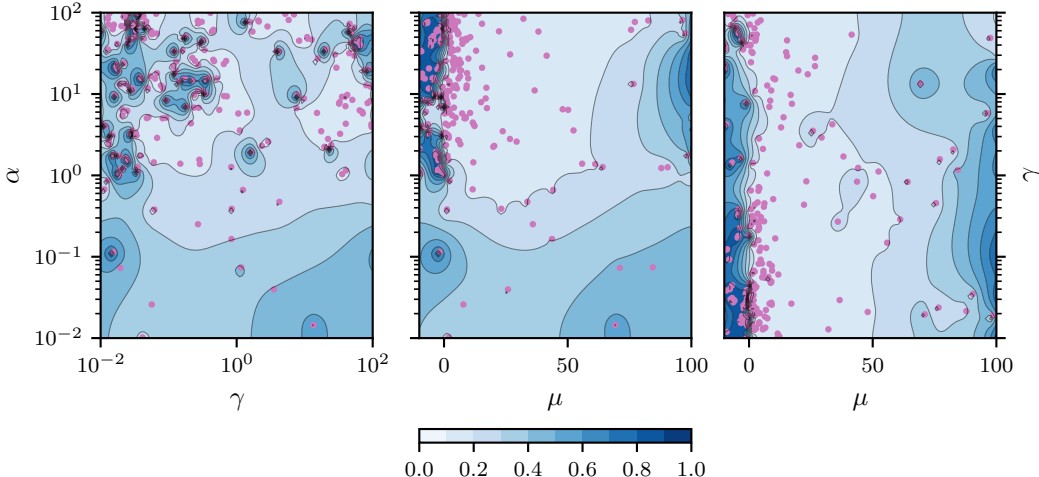

Figure 16: Landscapes of classification error for ResNet-20 model trained on CIFAR-10 with NAG-GS after projections onto $\alpha - \gamma$, $\alpha - \mu$ and $\gamma - \mu$ planes (from left to right). Hyperparameter optimization algorithm samples learning rate $\alpha$ from $[10^{-2}, 10^2]$, factor $\gamma$ from $[10^{-2}, 10^2]$, and factor $\mu$ from $[-10, 90]$. Hyperparameters $\alpha$ and $\gamma$ are sampled from log-uniform distribution, and hyperparameter $\mu$ is sampled from a uniform distribution.

Table 1: The comparison of a single step duration for different optimizers on RESNET-20 on CIFAR-10. ADAM-like optimizers have in twice larger state than SGD with momentum or NAG-GS.

| OPTIMIZER | MEAN, S | VARIANCE, S | REL. MEAN | REL. VARIANCE |
|---|---|---|---|---|
| SGD | 0.458 | 0.008 | 1.0 | 1.0 |
| NAG-GS | 1.648 | 0.045 | 3.6 | 5.5 |
| SGD-M | 3.374 | 0.042 | 7.4 | 5.2 |
| SGD-MW | 3.512 | 0.037 | 17.7 | 4.7 |
| ADAMW | 5.208 | 0.102 | 11.4 | 12.6 |
| ADAM | 7.919 | 0.169 | 17.3 | 20.8 |

## 3.2 IMPLEMENTATION DETAILS

In our work, we implemented NAG-GS in PyTorch Paszke et al. (2017) and JAX Bradbury et al. (2018); Babuschkin et al. (2020). Both implementations are used in our experiments and available online[2]. According to Algorithm 1, the size of the NAG-GS state equals to number of optimization parameters which makes NAG-GS comparable to SGD with momentum. It is worth noting that Adam-like optimizers have a twice larger state than NAG-GS. The arithmetic complexity of NAG-GS is linear $O(n)$ in the number of parameters. Table 1 shows a comparison of the computational efficiency of common optimizers used in practice. Although forward pass and gradient computations usually give the main contribution to the training step, there is a setting where the efficiency of gradient updates is important (e.g. batch size or a number of intermediate activations are small with respect to a number of parameters).

## 3.3 UPDATABLE SCALING FACTOR $\gamma$

According to the theory of NAG-GS optimizer presented in Section 2, the scaling factor $\gamma$ decays exponentially fast to $\mu$ and, in the case $\gamma_0 = \mu$, $\gamma$ remains constant along iterations. So, a natural question arises: is the update on $\gamma$ necessary? Our experiments confirm that scaling factor $\gamma$ should be updated accordingly to Algorithm 1, even in this highly non-convex setting, in order to get better metrics on test sets.

---

[2]https://github.com/user/nag-gs

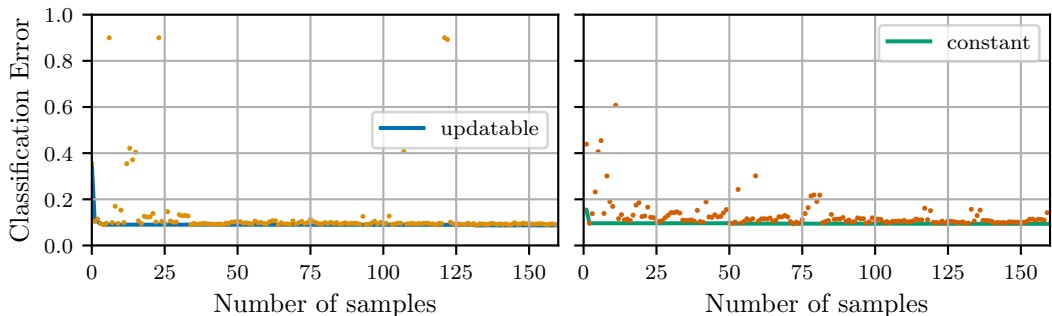

Figure 17: The best acc@1 on test set for updatable and fixed scaling factor $\gamma$ during hyperoptimization. NAG-GS with updatable $\gamma$ gives more frequently better results than the ones obtained with constant $\gamma$.

We use an experimental setup for ResNet-20 from Section 3.4 in the main text and search for hyperparameters for NAG-GS with updatable $\gamma$ and with constant one. Common hyper-optimization library OPTUNA Akiba et al. (2019) is used with a budget of 160 iterations to sample NAG-GS parameters. Figure 17 plots the evolution of the best score value along optimization time.

## 3.4 NON-CONVEXITY AND HESSIAN SPECTRUM

Theoretical analysis of NAG-GS highlights the importance of the smallest eigenvalue of the Hessian matrix for convex and strongly convex functions. Unfortunately, the objective functions usually considered for the training of neural networks are not convex. In this section, we try to address this issue. The smallest model in our experimental setup is ResNet-20. However, we cannot afford to compute exactly the Hessian matrix since ResNet-20 has almost 300k parameters. Instead, we use Hessian-vector product (HVP) $H(x)$ and apply matrix-free algorithms for finding the extreme eigenvalues. We estimate the extreme eigenvalues of the Hessian spectrum with power iterations (PI) along with Rayleigh quotient (RQ) Golub & van Loan (2013). PI is used to get a good initial vector which is used later in the optimization of RQ. In order to get a more useful initial vector for the estimation of the smallest eigenvalue, we apply the spectral shift $H(x) - \lambda_{\max}x$ and use the corresponding eigenvector.

Figure 20 shows the extreme eigenvalues of ResNet-20 Hessian at the end of each epoch for the batch size 256 in the same setup as in Section 3.4 in the main text. The largest eigenvalue is strictly positive while the smallest one is negative and usually oscillates around $-1$. It turns out that there is an island of hyperparameters in the vicinity of that $\mu$. We report that training ResNet-20 with hyperparameters included in this island gives good target metrics. The domain of negative momenta is non-conventional and not well understood, to the best of our knowledge. Moreover, there are no theoretical guarantees for NAG-GS in the non-convex case and negative $\mu$. However, Velikanov et al. (2022) reports the existence of regions of convergence for SGD with negative momentum, which supports our observations. The theoretical aspects of these observations will be studied in future work.

## 3.5 ADDITIONAL EXPERIMENTS WITH VIT

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

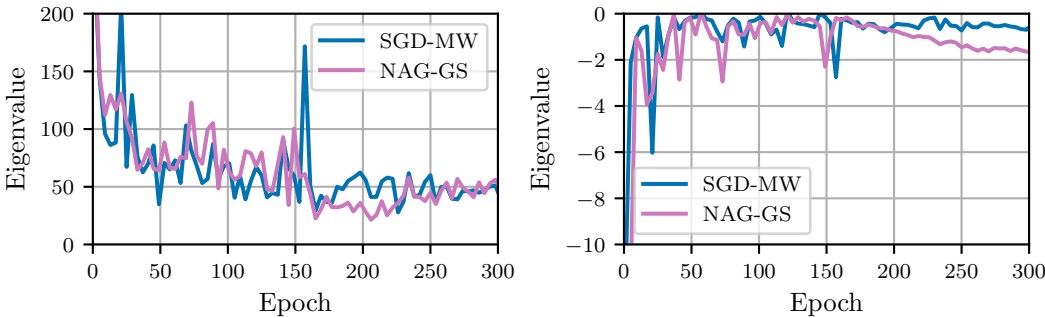

Figure 18: Evolution of the extreme eigenvalues (the largest and the smallest ones) during training RESNET-20 on CIFAR-10 with the NAG-GS optimizer.

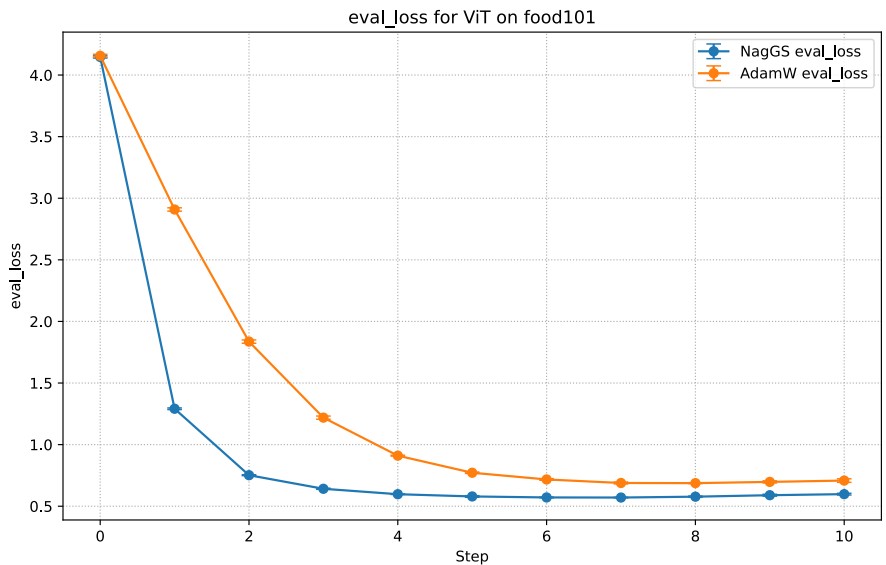

Figure 19: Comparison of NAG-GS and AdamW on the ViT training problem on food-101 dataset with the best hyperparameters found with the hyperparameter search on the portion of data. The number of experiments per method is 5. Mean values and standard errors of the evaluation loss are presented.

Chris Jones, Ross Hemsley, Tom Hennigan, Matteo Hessel, Shaobo Hou, Steven Kapturowski, Thomas Keck, Iurii Kemaev, Michael King, Markus Kunesch, Lena Martens, Hamza Merzic, Vladimir Mikulik, Tamara Norman, John Quan, George Papamakarios, Roman Ring, Francisco Ruiz, Alvaro Sanchez, Rosalia Schneider, Eren Sezener, Stephen Spencer, Srivatsan Srinivasan, Luyu Wang, Wojciech Stokowiec, and Fabio Viola. The DeepMind JAX Ecosystem, 2020. URL http://github.com/deepmind.

Aleksandar Botev, Hippolyt Ritter, and David Barber. Practical Gauss-Newton optimisation for deep learning. In *Proceedings of the 34th International Conference on Machine Learning*, volume 70 of *Proceedings of Machine Learning Research*, pp. 557–565. PMLR, 06–11 Aug 2017.

James Bradbury, Roy Frostig, Peter Hawkins, Matthew James Johnson, Chris Leary, Dougal Maclaurin, George Necula, Adam Paszke, Jake VanderPlas, Skye Wanderman-Milne, and Qiao Zhang. JAX: Composable Transformations of Python+NumPy Programs, 2018. URL http://github.com/google/jax.

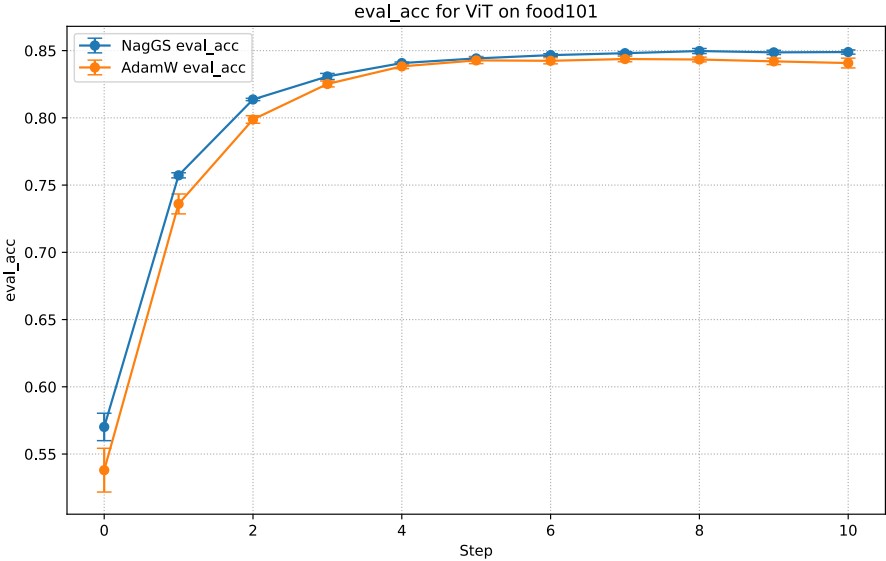

Figure 20: Comparison of NAG-GS and AdamW on the ViT training problem on food-101 dataset with the best hyperparameters found with the hyperparameter search on the portion of data. The number of experiments per method is 5. Mean values and standard errors of the evaluation accuracy are presented.

Sinho Chewi, Thibaut Le Gouic, Chen Lu, Tyler Maunu, and Philippe Rigollet. Svgd as a kernelized wasserstein gradient flow of the chi-squared divergence. *Advances in Neural Information Processing Systems*, 33:2098–2109, 2020.

Germund G Dahlquist. A special stability problem for linear multistep methods. *BIT Numerical Mathematics*, 3(1):27–43, 1963.

Gene H. Golub and Charles F. van Loan. *Matrix Computations*. JHU press, 2013.

Stanisław Jastrzębski, Zachary Kenton, Devansh Arpit, Nicolas Ballas, Asja Fischer, Yoshua Bengio, and Amos Storkey. Three factors influencing minima in sgd, 2018.

Nitish Shirish Keskar, Dheevatsa Mudigere, Jorge Nocedal, Mikhail Smelyanskiy, and Ping Tak Peter Tang. On large-batch training for deep learning: Generalization gap and sharp minima, 2017.

Marc Lambert, Sinho Chewi, Francis Bach, Silvère Bonnabel, and Philippe Rigollet. Variational inference via wasserstein gradient flows. *arXiv preprint arXiv:2205.15902*, 2022.

Jonas Latz. Analysis of stochastic gradient descent in continuous time. *Statistics and Computing*, 31 (4):1–25, 2021.

Kenneth Levenberg. A method for the solution of certain non-linear problems in least squares. *Quart. Appl. Math.*, 2, 1944.

Hao Luo and Long Chen. From differential equation solvers to accelerated first-order methods for convex optimization. *Mathematical Programming*, pp. 1–47, 2021.

Donald W. Marquardt. An algorithm for least-squares estimation of nonlinear parameters. *Journal of the Society for Industrial and Applied Mathematics*, 11(2):431–441, 1963.

Gisiro Maruyama. Continuous markov processes and stochastic equations. *Rendiconti del Circolo Matematico di Palermo*, 4(1):48–90, 1955.

Yurii Nesterov. A method of solving a convex programming problem with convergence rate $\mathcal{O}\left(1/k^2\right)$. In *Doklady Akademii Nauk*, volume 269, pp. 543–547. Russian Academy of Sciences, 1983.

Yurii Nesterov. *Lectures on Convex optimization*, volume 137. Springer Optimization and Its Applications, 2018.

Adam Paszke, Sam Gross, Soumith Chintala, Gregory Chanan, Edward Yang, Zachary DeVito, Zeming Lin, Alban Desmaison, Luca Antiga, and Adam Lerer. Automatic Differentiation in PyTorch, 2017.

Fernando Pérez-Cruz. Kullback-leibler divergence estimation of continuous distributions. In *2008 IEEE international symposium on information theory*, pp. 1666–1670. IEEE, 2008.

Boris T Polyak. Some methods of speeding up the convergence of iteration methods. *Ussr computational mathematics and mathematical physics*, 4(5):1–17, 1964.

G.O. Roberts and O. Stramer. Langevin diffusions and metropolis-hastings algorithms. *Methodology And Computing In Applied Probability*, 4:337–357, 2002.

Bryan Van Scoy, Randy A Freeman, and Kevin M Lynch. The fastest known globally convergent first-order method for minimizing strongly convex functions. *IEEE Control Systems Letters*, 2(1): 49–54, 2017.

Maksim Velikanov, Denis Kuznedelev, and Dmitry Yarotsky. A view of mini-batch sgd via generating functions: conditions of convergence, phase transitions, benefit from negative momenta. 2022. doi: 10.48550/arxiv.2206.11124. URL https://arxiv.org/abs/2206.11124.

Matthew D. Zeiler. Adadelta: An adaptive learning rate method, 2012.