# OpenReview forum: "NAG-GS: Semi-Implicit, Accelerated and Robust Stochastic Optimizer"
_ICLR.cc/2024/Conference — Submitted to ICLR 2024_

### Official Review · Reviewer_f3XV · 2023-10-26

**Soundness:** 2 fair
**Presentation:** 2 fair
**Contribution:** 2 fair
**Rating:** 3
**Confidence:** 3

**Summary:**

This paper proposes a novel, robust and accelerated stochastic optimizer based on combining Nesterov's AGD and semi-implicit Gauss-Seidel method to obtain an iterative scheme for the optimization. Convergence of this algorithm in the quadratic case is proved and numerical experiments are demonstrated for logistic regression model, ResNet-20, VGG-11 and Transformers.

**Strengths:**

proposed a novel stochastic optimizer by combining Nesterov AGD with Gauss-Seidel semi-implicit method

**Weaknesses:**

(1) it seems the contribution is relatively moderate by just combining Nesterov with Gauss-Seidel
(2) some of the writings are sloppy. Example: page 4, line 2, two "either" appeared

**Questions:**

N/A

---

> ### Author Response · Authors · 2023-11-23
>
> Dear reviewer f3XV,
>
> thanks for your comment on our manuscript!
> Below we have highlighted the main contribution and novelty of our study:
> 1) We introduced Gauss-Seidel discretization of the second-order ODE related to the accelerated gradient method in the stochastic setup
> 2) We demonstrated numerically the expected robustness w.r.t. the large step size. Note that at the end of Appendix 1, we detail an additional yet important motivation for proposing a solver capable of handling a broad range of step size values. Specifically, allowing for a larger ratio $\alpha/b$ (b as the mini-batch size) increases the likelihood of converging to wider local minima, ultimately enhancing the generalization performance of the trained model.
> 3) We proved the convergence theorem in the quadratic case and obtained the optimal learning rate that maximizes the contraction and therefore corresponds to the fastest convergence
> 4) We presented an extensive benchmarking of NAG-GS in the stochastic setup, where different tasks from computer vision and natural language processing domains are considered. In the presented experimental results, NAG-GS shows superior or on-par performance compared with competitors.
>
> In addition, we have carefully proofread the manuscript and fixed typos.

---

### Official Review · Reviewer_13Pv · 2023-10-28

**Soundness:** 3 good
**Presentation:** 2 fair
**Contribution:** 2 fair
**Rating:** 5
**Confidence:** 4

**Summary:**

This work proposes a stochastic method through applying implicit Gauss-seidel splitting method on the continuous-time ODE mimicing the trajectory of Nesterov’s accelerated method. They prove the convergence of their algorithm for the special case of strongly-convex quadratic functions in Theorem 1. They extended their algorithm to non-convex functions heuristically based on the step-size they found for the special case of strongly convex quadratic functions. Several experiments were conducted to evaluate the performance of their method.

**Strengths:**

1- Connecting theoretical findings with practical applications and implementations.

2- One step toward practical implementations through considering stochastic extension of prior art in [Luo & Chen (2021)].

3- Text is smooth and easy to read.

**Weaknesses:**

1- Related work is very imited. The idea of analyzing accelerated methods through their continuous time perspective has been around for quite some time (since [Su, et. al (2014)] or even before that by [Alvarez & Attouch (2001)] and most of the related works mentioned deal with intrepretations of deterministic methods. It makes more sense to focus on works that see stochastic accelerated methods through the lens of ODEs since these are more related to the proposed research.

2- The theoretical analysis is bounded to quadratic case. This is not mentioned accurately in the contribution. Specifically the second main contribution is: **We analyze the properties of the proposed method both theoretically and empirically;**.

3- The introduction is not really an introduction of this work. By just reading the introduction, it is not possible to get an accurate idea of what differences this work has with any other work in "stochastic optimization algorithms".

4- "The Preliminaries" is a mixture of background, related work and notations. Here, more organization might improve readability.

5- Gaus-Seidel splitting used here is not a novel idea in discretizing ODEs for acceleration as it was previously discussed in [Luo & Chen (2021)].


References

Alvarez, F., Attouch, H. An Inertial Proximal Method for Maximal Monotone Operators via Discretization of a Nonlinear Oscillator with Damping. Set-Valued Analysis 9, 3–11 (2001).

**Questions:**

1- What is $\mathcal R(\lambda)$ under (8)? Does it extract the real part of $\lambda$?

2- Have you tried the predictor-corrector method in [Luo & Chen (2021)] and extend it to the stochastic case (like the way you did for the semi-implicit GS)?

3- [Shi, et al. (2019)] showed that semi-implicit Euler discretization of a “high-resolution ODE” exactly recovers the NAG algorithm. This is not the case for other ODEs like the one in  [Su, et. al (2014)] or [Luo & Chen (2021)], unless some corrections are made. Do you think applying semi-implicit GS on high-resolution ODEs (combined with stochastic gradients) can lead to an even better algorithm?

---

> ### Author Response · Authors · 2023-11-23
>
> Dear reviewer 13Pv, thanks for your thorough comments and remarks! Below we provide requested clarifications and address the mentioned issues. The main text is also revised respectively.
>
> Weaknesses
>
> * The theoretical analysis is bounded to quadratic case. This is not mentioned accurately in the contribution. Specifically the second main contribution is: We analyze the properties of the proposed method both theoretically and empirically;.
>
> Answer: Thank you for your comment, we have clarified our theoretical contribution for the quadratic case in the main text.
>
> * The introduction is not really an introduction of this work. By just reading the introduction, it is not possible to get an accurate idea of what differences this work has with any other work in "stochastic optimization algorithms".
>
> Answer: We place in the Introduction section the main contributions and the organization of the manuscript that provides step-by-step plan on how we analyze the proposed dmethod theoretically and empirically.
>
> * "The Preliminaries" is a mixture of background, related work and notations. Here, more organization might improve readability.
>
> Answer: thanks for the suggestions! We have slightly revised this section. The main intention here was to briefly introduce the reader to the used concept of ODE-inspired optimizers and clarify the basics necessary for the further development of NAG-GS.
>
> * Gaus-Seidel splitting used here is not a novel idea in discretizing ODEs for acceleration as it was previously discussed in [Luo & Chen (2021)].
>
> Answer: Thank you for your valuable comment. It's acknowledged that deterministic Gauss-Seidel splitting in the context of accelerating SDEs was previously discussed by Luo & Chen (2021). In our work, we introduce novel contributions, particularly in the quadratic case:
> Extension of Deterministic Results: We provide a detailed analysis of the optimal step size computation, which exceeds the limit proposed by Luo and Chen (refer to Lemma 1 in our paper). Additionally, we identify a scenario (μ > L/2) where we can freely choose the step size, offering a new perspective.
>
> Stochastic Setting under Gaussian Noise: We extend our approach to the stochastic setting under a Gaussian assumption for the gradient noise, introducing a family of accelerated SDEs (Equation 11). We prove the asymptotic convergence of NAG-GS with the same optimal step size as in the deterministic case. We provide optimal choice for the step size leading to the highest possible contraction rate when aiming at minimizing a strongly convex quadratic form. Furthermore, our numerical experiments demonstrate the expected robustness concerning large step sizes across a diverse range of problems.
>
> Additionally, in Appendix 1.2, we explore a fully implicit version. Our findings reveal that, in the stochastic case for quadratic minimization, we can select any step size, mitigating the noise effect. This implies that the spreading of particles can be arbitrarily close to the arg min, showcasing a notable result.
>
> References:
> Alvarez, F., Attouch, H. An Inertial Proximal Method for Maximal Monotone Operators via Discretization of a Nonlinear Oscillator with Damping. Set-Valued Analysis 9, 3–11 (2001).

---

> ### Author Response · Authors · 2023-11-23
>
> The second part of the comment
>
> High-Resolution ODE Framework in a nutshell:
> Introduced by Shi et al. (2019), the framework distinguishes between NAG-SC and Polyak's heavy-ball method, revealing a unique "gradient correction" term for NAG-SC. Incorporating O(√s) terms in the limiting process yields high-resolution ODEs, offering more accurate continuous-time representations.
>
> ODE Formulations:
> Here-under, we recall two key ODEs which are presented for the heavy-ball method and NAG-SC, highlighting distinctive terms:
>
> The high-resolution ODE for the heavy-ball method:
> $$  \ddot{x}(t) + 2 \sqrt{\mu} \dot{x}(t) + (1+\sqrt{\mu s}) \nabla f(x(t)) = 0  $$
>
> The high-resolution ODE for “NAG-SC”:
> $$      \ddot{x}(t) + 2\sqrt{\mu} \dot{x}(t) + \sqrt{s}  \nabla^2 f(x(t)) \dot{x}(t) +   (1+\sqrt{\mu s}) \nabla f(x(t)) = 0    $$
>
> According to Shi et al. (2019), these ODEs, with specific initial conditions, provide a better characterization of accelerated methods in continuous time.
>
> Numerical Stability and Step Size:
> Section 3.3 of Shi et al. (2019) addresses numerical stability, deriving bounds for step size (s) with a forward Euler scheme.
> A comparison reveals that the Hessian makes the forward Euler scheme for NAG-SC ODE numerically stable with a larger step size (s = O(1/L)).
> -> Clearly here, there is a strong motivation for a semi-implicit approach to ensure a larger step size than pure explicit discretizations.
>
> However, we would like to raise the following points:
>
> Considerations in Noisy Settings: The previous considerations are relevant in a deterministic setup, but we need for further analysis in noisy gradients. A detailed examination, especially for quadratic forms minimization in noisy settings, could provide insights into step size choices. Furthermore, simple numerical tests as the ones done in Appendix 1.1.5 could be helpful to gain some insights.
>
> Transformation and Condition Number: In our paper, we considered one transformation to derive accelerated ODE, which relies on the embedding of matrix A into a 2 × 2 block matrix G, to reduce the condition number and enable the use of larger step sizes. We do not know if, in the end, using semi-implicit discretization for NAG-SC high-res ODE would offer a similar range for step size choices.
>
> Gradient Corrections and Second-Order Information: In Shi et al. (2019), authors highlight the importance of the good estimation of Hessian-vector products (the gradient-correction terms) in NAG-SC high-res ODE, imposing step size s is not too big, with a claim that even small gradient corrections have a fundamental effect on the scheme's behavior. Moreover, their gradient corrections bring in second-order information from the objective function, which could be an interesting way to develop efficient methods using higher-order methods without explicitly building the Hessian and inverting as it is the case for Newton-steps based methods. Furthermore, in practice, efficient routines exist to accurately compute such Hessian-vector products even in large-scale problems.
> But again, in the stochastic setting, for instance using stochastic estimates of the Hessian, or stochastic estimates of the Hessian-vector products on a mini-batch, we do not know yet how this approach could behave overall and if stochastic estimates of gradient-corrections terms still bring something. Among many interrogations, we would also be interested to see what would happen close to stationarity. Using additional information on the function to minimize, such as the Hessian, may indeed speed up the converge of the method in the early stages (killing the gradient faster:), that is when the noise is low compared the gradient in terms of “norm”, but what happens when the noise counterpart becomes in the same order of magnitude than the gradient ?
>
> We are eager to further discuss these ideas with the reviewer and appreciate the inspiration provided.
>
> LINKS for the additional experiments
>  http://bitly.ws/wSbz
>  http://bitly.ws/wSbC
>  http://bitly.ws/wSbH

---

### Official Review · Reviewer_nCbP · 2023-11-02

**Soundness:** 3 good
**Presentation:** 3 good
**Contribution:** 2 fair
**Rating:** 5
**Confidence:** 4

**Summary:**

This paper introduces a novel stochastic optimizer called NAG-GS, which combines elements of Nesterov-like Stochastic Differential Equation (SDE) acceleration and semi-implicit Gauss-Seidel type discretization. The method's convergence and stability are extensively analyzed, particularly in the context of minimizing quadratic functions. The authors determine an optimal learning rate that balances convergence speed and stability by considering various hyperparameters. NAG-GS is shown to be competitive with other state-of-the-art methods like momentum SGD with weight decay and AdamW when applied to various machine learning models, including logistic regression, residual networks, Transformers, and Vision Transformers across different benchmark datasets.

**Strengths:**

- The NAG-GS is derived from an accelerated Stochastic Differential Equation (SDE) using its semi-implicit Gauss-Seidel type discretization, which is interesting.

- The convergence analysis for the quadratic case is comprehensive.

**Weaknesses:**

- The discussion of NAG-GS with other similar methods is insufficient. For example, Is NAG-GS faster than Polyak's momentum method for solving quadratic objectives? How does it compare with other variants of NAG, such as Triple momentum method [1, 2], and ITEM [3]? It is not clear what is the key benefit of NAG-GS in its original setting.

- The improvement in the neural network experiments seems marginal. No deviation statistics is provided in the empirical results.

- A minor point: As one of the key features of NAG-GS, the derived optimal learning rate should be mentioned and discussed in the main text.

=================== After Rebuttal ======================

Thanks the authors for their revision and detailed feedback. The contribution is much clearer and I have increased the score to 5. In my humble opinion, the contribution (theoretical and empirical) is still not sufficient for acceptance as pointed out by other reviewers.

=====================================================


[1] Van Scoy, B., Freeman, R. A., & Lynch, K. M. (2017). The fastest known globally convergent first-order method for minimizing strongly convex functions. IEEE Control Systems Letters, 2(1), 49-54.


[2] Zhou, K., So, A. M. C., & Cheng, J. (2020). Boosting first-order methods by shifting objective: new schemes with faster worst-case rates. Advances in Neural Information Processing Systems, 33, 15405-15416.


[3] Taylor, A., Drori, Y. An optimal gradient method for smooth strongly convex minimization. Math. Program. 199, 557–594 (2023).

**Questions:**

See weaknesses.

---

> ### Author Response · Authors · 2023-11-23
>
> Dear Reviewer nCbP,
>
> Thank you for your insightful feedback on our manuscript. We appreciate the opportunity to address the concerns raised and have made several amendments to our paper to reflect these points.
>
> 1. **Comparison with Other Methods:**
>    - We recognize the importance of comparing NAG-GS with existing methods like Polyak's momentum method and Triple momentum method. To this end, We have checked the mentioned references and updated Section 4. Both ITEM and Triple momentum do not study the robustness w.r.t, the large step size and consider only deterministic setup. The incorporting these methods in the ODE-inspired setup is the interesting future work. This comparison highlights the unique advantages of NAG-GS, particularly in its original setting.
> We added additional figures 15 and 16 to the section "Numerical tests for quadratic case" in supplementary materials, which illustrates the fastest asymptotic convergence rate of the NAG-GS for the considered quadratic problems. The figures are also available here [15](https://freeimage.host/i/JoRuOKv) and [16](https://freeimage.host/i/JoRuwiJ).
> To illustrate the comparison the NAG-GS with Polyak’s momentum metod we also have updated Figure 1.
>
> 2. **Experimental Improvement and Deviation Statistics:**
>    - We acknowledge that the improvement in neural network experiments appeared marginal and lacked deviation statistics. To address this, we have now included a set of experiments of NAG-GS and AdamW on the ViT training problem on food-101 dataset with the best hyperparameters found with the hyperparameter search on the portion of data. The number of experiments per method is 5. Mean values and standard errors of the evaluation loss are presented. . The results are included in the section Additional experiments with ViT in Appendix. [Link for eval accuracy graph](https://freeimage.host/i/JoRAykN) [Link for eval loss graph](https://freeimage.host/i/JoRAp7p)
>
> 3. **Optimal Learning Rate Discussion:**
>    - Based on your suggestion, we have included Remark 2 in the main text, which explicitly discusses the derived optimal learning rate for NAG-GS. This addition emphasizes one of the key features of NAG-GS and its relevance in the context of gradient methods.
>
> 4. **Further Empirical Comparisons and Future Work:**
>    - We have also updated Figure 1 to illustrate a direct comparison of NAG-GS with Polyak’s momentum method. This visual representation aids in understanding the relative performance of these methods.
>    - Recognizing the need for broader empirical evidence, we have conducted additional experiments using Vision Transformers (ViT) and provided results with error bars for a more nuanced evaluation.
>    - We see the integration of methods like ITEM and Triple momentum in an ODE-inspired setup as an exciting avenue for future research and have mentioned this as part of our conclusion and future work section.
>
> We believe these revisions and additions significantly strengthen our manuscript and address the concerns you have raised. We are grateful for the opportunity to enhance our work based on your feedback and hope that these changes meet your expectations for a robust and comprehensive analysis.

---

### Official Review · Reviewer_MgiM · 2023-11-07

**Soundness:** 4 excellent
**Presentation:** 4 excellent
**Contribution:** 4 excellent
**Rating:** 6
**Confidence:** 3

**Summary:**

This paper proposes a new accelerated optimization algorithm using gauss-siedel discretization of a randomization version of the recent spectral lifting technique of Luo and Chuo 2021

**Strengths:**

The paper is largely well written with minor polishing still required. The method seems sound from a few empirical and numerical experiments conducted by the authors, and achieves competitive performance.

**Weaknesses:**

The main weakness is that the convergence analysis is only for the quadratic case, and the convergence itself as stated in theorem 1 is a weak statement with only asymptotic convergence. I did not go through the entire proof, but i can understand that the analysis for general f (replacing Ax with grad f  for the method) is non-trivial, since the "lifted" spectrum needs to be bounded effectively. It seems one also requires prior knowledge of \mu for the algorithm which is very limiting, and the convergence analysis also is only valid for strongly convex cases.

However, the empirical results are quite strong which warrants that the community should know about this paper.

There are some typos etc that need to be fixed.

**Questions:**

Algorithm 1 requires prior knowledge of \mu. How did you set up the algorithm practically for non-convex losses ?

Mostly minor writing stuff:

Background: Please explain the definition of A-stable. If you have gone as far to explain the discretization process itself, adding A-stable for reading not familiar with it would benefit from it.

What is \mathcal{R}(\lambda) below (8) ?

“However, this requires to either solve a linear system either.” --> “However, this requires to either solve a linear system or.”

Please mention what is the baseline SGD-MW before using it in table 1. Is there a reason Adam or a variant of it was considered as a baseline for Resnet20 ?

What is the accelerated gradient descent baseline in Figure 1 ?

“But still, it is expected that an explicit scheme closer to the implicit Euler method will have good stability with a larger step size than the one offered by a forward Euler method. “ – why ?

Can the authors provide some future work directions ? convergence analysis ?

---

> ### Author Response · Authors · 2023-11-23
>
> Dear Reviewer MgiM,
>
> Thank you for your insightful feedback on our paper. Below we provide detailed responses to the mentioned weaknesses and answer the questions.
>
> **Convergence Analysis and Empirical Results**:
> We acknowledge that extending analysis to more general functions introduces additional complexities and is the question for the further research. Regarding the requirement of prior knowledge of $\mu$, we approached this by aligning the weight decay parameter used in standard optimizers (like SGD, Adam, etc.) with the strongly convex constant $\mu$. Consequently, in our experiments, we set the constant for weight decay in competitor algorithms and for the $\mu$ in NAG-GS correspondingly.
>
> **Algorithm 1 and Non-Convex Losses**:
> In practical applications for non-convex losses, we derived the $\mu$ parameter from the weight decay settings commonly employed in standard optimizers. This approach allowed us to integrate our method into existing frameworks without additional complexity. For some experiments (for example, comparing AdamW and NAG-GS on the ViT experiment on food101) we initially performed hyperparameter search for all hyperparameters for the methods on the portion of the dataset and compared them with the best choice for both.
>
> **Clarifications and Typos**:
> We have made several updates to our manuscript to clarify points you've raised:
> - **Definition of A-stability**: We added a high-level definition and reference in the main text and detailed insights in Appendix 1.3.
> - **Clarification of $\mathcal{R}(\lambda)$**: We now specify that $\mathcal{R}(\lambda)$ refers to the real part of $\lambda$.
> - **SGD-MW**: We clarified that this stands for Stochastic Gradient Descent with Momentum and Weight decay, and added this explanation before its first use in the text.
> - **Baseline Algorithms**: The choice of SGD-MW as a baseline for ResNet-20 was due to its comparable memory footprint to our NAG-GS. In contrast, Adam requires two state vectors, instead of one in case of NAG-GS ad SGD-MW. Additionally, the accelerated gradient descent in Figure 1 references the Nesterov accelerated method variant presented in Su et al. (2014).
> - **Typographical Corrections**: We corrected the typo you pointed out and carefully reviewed the manuscript for any other errors.
>
> **Additional Insights on Stability and Step Size**:
> $A$-stable methods, exemplified by the implicit Euler scheme, exhibit a notable advantage in that they impose no inherent limitations on the integration step (or step size). On the other hand, fully explicit methods, like the forward Euler scheme (which essentially translates to gradient steps), necessitate certain bounds on integration steps to ensure stability and asymptotic convergence. To strike a balance, we explore a semi-implicit scheme, aiming to permit a broader range of values for the learning rate (or step size) compared to pure explicit discretization. Importantly, this is achieved without incurring the computational cost associated with the fully-implicit approach.
>
> For a comprehensive understanding of this aspect, please refer to the additional insights provided in the new Appendix 1.3 and Appendix 1.2, the latter of which is dedicated to the fully implicit discretization of the proposed accelerated SDE (Equation 11).
>
> In the specific case of the quadratic scenario with $\gamma=\mu$, we can theoretically establish that NAG-GS allows for an optimal step size (($\alpha = \frac{2 \mu + 2 \sqrt{\mu L}}{L-\mu}$)), as detailed in Theorem 1 and Remark 2. This optimal step size is strictly higher than the maximum step size of $\frac{2}{L}$ (itself higher than $\frac{2}{\mu + L}$) permitted by the traditional Gradient Descent (GD) method for convex (or $\mu$-convex) $L$-smooth functions in a deterministic setting, under certain conditions. These conditions include $L > 0$ and ( $\frac{L}{(L+1)^2} < \mu < L$). Empirically, our findings demonstrate that NAG-GS facilitates the utilization of a broad range of step sizes, proving effective across a diverse set of problems when compared to AdamW and SGD with Momentum.
>
> Last but not least, in the end of Appendix 1, we detail an additional yet important motivation of proposing a solver capable of handling a broad range of step size values. Specifically, allowing for a larger ratio $\frac{\alpha}{b}$ ($b$ as the mini-batch size) increases the likelihood of converging to wider local minima, ultimately enhancing the generalization performance of the trained model.
>
> **Future Directions**:
> Regarding future work, we have expanded Section 5 of our manuscript to outline potential directions. These include extending our convergence analysis to broader classes of functions and investigating the applicability of our approach to different machine learning models and problem domains, especially for gradient noise with unbounded variance.
>
> Thank you once again for your constructive comments, which have greatly contributed to improving the quality of our paper.

---

### Meta-Review · Area_Chair_SXqb · 2023-12-19

**Metareview:**

This work considers the implicit Gauss-seidel splitting method on the continuous-time ODE mimicing the trajectory of Nesterov’s accelerated method which was previously considered by Luo and Chen 2021. They prove the convergence of the algorithm for the special case of strongly-convex quadratic functions in Theorem 1 with stochastic noise. The further show that this method is stable at a higher learning rate on quadratics as compared to polyak's momentum and a version of AGD. The non-convex extensions are heuristically based on the step-size they found for the special case of strongly convex quadratic functions. They conducted experiments to verify their methods efficacy on neural networks.

The reviewers found the contribution of the paper clear and concise but for the following reasons they deemed the contributions to be insufficient.

1. Overall the idea proposed by the paper is not novel. The algorithm and the setup was proposed by Luo and Chen and the paper's main contribution is a stochastic analysis of the method.
2. The paper's writing raised several questions by the reviewers and the key points of improvements pointed out were contextualizing the results and contributions correctly with appropriate comparisons with existing work, the presentation of the method itself and the presentation of the experiments with further clarity required in terms of hyperparameters and robustness.
3. The issue of the contributions being limited to quadratics was also a limitation.

Overall the paper's contributions are in a solid direction and I believe that the paper is a borderline paper which can become a solid paper after taking into account the issues and questions raised by the reviewers. I strongly encourage the authors to take these into account.

**Justification For Why Not Higher Score:**

Several points raised by the reviewers included limited scope of results, limited novelty, room for significantly better exposition and clarity. Please see details in meta review.

**Justification For Why Not Lower Score:**

NA

---

### Decision · Program_Chairs · 2024-01-16

Reject